# Think-on-Graph 3.0: Efficient and Adaptive LLM Reasoning on Heterogeneous Graphs via Multi-Agent Dual-Evolving Context Retrieval

## Abstract

Retrieval-Augmented Generation (RAG) and Graph-based RAG has become the important paradigm for enhancing Large Language Models (LLMs) with external knowledge. However, existing approaches face a fundamental trade-off. While graph-based methods are inherently dependent on high-quality graph structures, they face significant practical constraints: manually constructed knowledge graphs are prohibitively expensive to scale, while automatically extracted graphs from corpora are limited by the performance of the underlying LLM extractors, especially when using smaller, local-deployed models. This paper presents Think-on-Graph 3.0 (ToG-3), a novel framework that introduces Multi-Agent Context Evolution and Retrieval (MACER) mechanism to overcome these limitations. Our core innovation is the dynamic construction and refinement of a Chunk-Triplets-Community heterogeneous graph index, which pioneeringly incorporates a dual-evolution mechanism of Evolving Query and Evolving Sub-Graph for precise evidence retrieval. This approach addresses a critical limitation of prior Graph-based RAG methods, which typically construct a static graph index in a single pass without adapting to the actual query. A multi-agent system, comprising Constructor, Retriever, Reflector, and Responder agents, collaboratively engages in an iterative process of evidence retrieval, answer generation, sufficiency reflection, and, crucially, evolving query and subgraph. This dual-evolving multi-agent system allows ToG-3 to adaptively build a targeted graph index during reasoning, mitigating the inherent drawbacks of static, one-time graph construction and enabling deep, precise reasoning even with lightweight LLMs. Extensive experiments demonstrate that ToG-3 outperforms compared baselines on both deep and broad reasoning benchmarks, and ablation studies confirm the efficacy of the components of MACER framework.

## 1 Introduction

The rapid advancement of both commercial (OpenAI, 2025; AI, 2025a; Comanici et al., 2025) and open-source Large Language Models (LLMs) (Yang et al., 2025; AI, 2025b; Liu et al., 2024; Zeng et al., 2025; Gan et al., 2023) has significantly enhanced the accessibility of generative AI capabilities for both end-users and developers. Notably, open-source models play a crucial role in enabling AI applications in offline environments. However, current LLMs still face notable limitations, including issues with factual hallucinations and inadequate performance in complex reasoning tasks. Retrieval-augmented generation (RAG) (Gao et al., 2023) has become a popular method for grounding Large Language Models (LLMs) with external knowledge, addressing issues like knowledge cutoff and hallucination. While traditional RAG systems rely on vector similarity to retrieve relevant text chunks, they often struggle with complex reasoning tasks that require integrating information across multiple documents or understanding structural relationships between entities. To address the above limitations, recent advancements have explored using Knowledge Graphs (KGs) or extracted Graph using LLMs to represent and retrieve structured information. ToG (Sun et al., 2023; Ma et al., 2024) pioneered an iterative hybrid RAG framework that tightly couples text and KGs retrieval, though their approach relies on pre-existing structured KGs such as Freebase and Wikidata. On the other hand, methods like GraphRAG (Edge et al., 2024) and LightRAG (Guo et al., 2024) address this issue by constructing a graph directly from the input documents. They create an entity-based graph to enhance information retrieval and summarization. However, as shown in Figure 1, the quality of the generated graph is highly dependent on the LLM's ability to accurately extract entities and

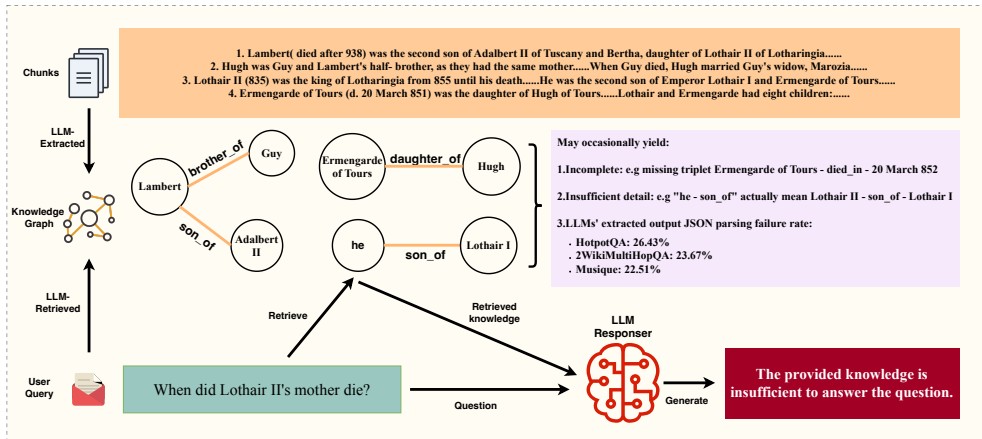

**Figure 1:** Performance Limitations of Graph-Based RAG systems under Resource-Constrained and Locally-Deployed Scenarios. In such scenarios, developers typically adopt open-source models such as Llama or Qwen as the backbone LLMs. Limitations like incomplete extracted triplets, insufficient extraction details and parsing failure may lead to insufficient knowledge provision, ultimately resulting in failure to adequately answer the query.

relationships, which can be a bottleneck for lightweight models like Qwen2.5-7B∼72B (Yang et al., 2024), which is broadly deployed in private and offline environments. Moreover, these methods often separate the handling of local and global questions.

To overcome these limitations, we introduce **Think-on-Graph 3.0** (ToG-3), a new RAG framework that integrates the strengths of both paradigms. Our core contribution lies in the introduction of a novel Chunk-Triplets-Community heterogeneous graph architecture and a novel MACER (Multi-Agent Context Evolution and Retrieval) mechanism, which pioneeringly incorporates a dual-evolution mechanism of **Evolving Query** and **Evolving Sub-Graph** for precise evidence retrieval. Figure 2 illustrates the key distinctions between ToG-3 and classical RAG paradigms such as NaiveRAG and GraphRAG. ToG-3 introduces a novel dual-evolution mechanism—comprising Evolving Query and Evolving Subgraph—that dynamically refines both the query representation and the graph structure in an iterative manner. This approach addresses a critical limitation of prior RAG methods, which typically construct a static graph index in a single pass without adapting to the actual query. The framework is particularly suited for resource-constrained and on-premises deployment scenarios, where lightweight open-source LLMs (e.g., Llama or Qwen) are often employed as the backbone of the RAG system.

Extensive experiments on complex multi-hop reasoning benchmarks demonstrate that our method achieves the highest average Exact Match and F1 scores on HotpotQA, 2WikiMultihopQA, and Musique. For broad reasoning tasks, ToG-3 also achieves remarkable win rates over baselines across comprehensiveness, diversity, and empowerment dimensions.

Our key contributions are summarized as follows:

1. We propose **MACER** (Multi-Agent Context Evolution and Retrieval), a novel multi-agent framework that introduces a dual-evolution mechanism integrating Evolving Query and Evolving Sub-Graph within graph-based RAG. This design significantly enhances retrieval performance and complex reasoning capabilities, especially when using lightweight open-source LLMs as the backbone of the RAG system.

2. We present **ToG-3**, a unified reasoning system that effectively combines the complementary advantages of prior graph-based and ToG methods through a Chunk–Triplet–Community Heterogeneous Graph Index and a Dual-Evolving Context Retrieval Loop Process.

3. We conduct extensive experiments on both **Deep and Broad Reasoning Tasks**, demonstrating that our approach consistently supports multi-hop inference and large-scale contextual integration, achieving competitive results across diverse benchmarks.

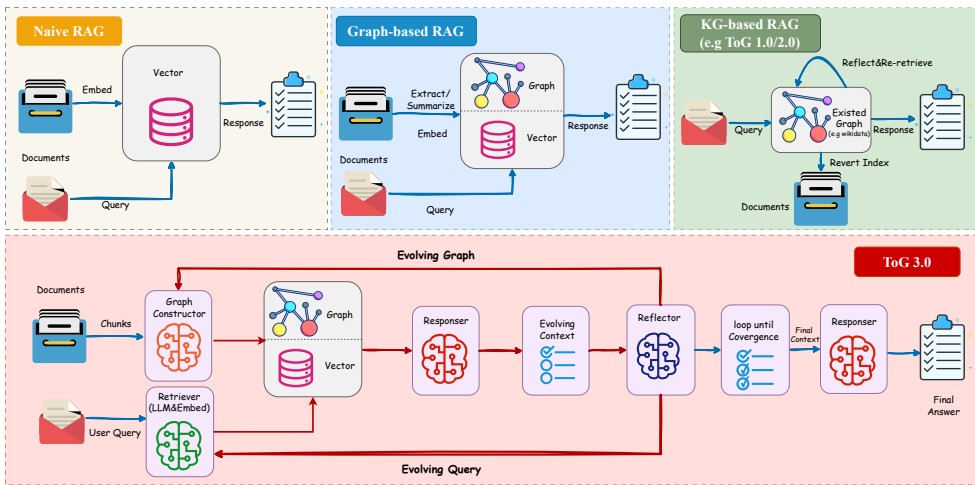

**Figure 2: Evolution of Retrieval-Augmented Generation Paradigms.** (**a**) Naive RAG embeds raw documents and performs single-shot retrieval. (**b**) Graph-based RAG pre-builds a static graph once and retrieves from it. (**c**) ToG-3 introduces a *four-agent* loop—Retriever, Constructor, Reflector, Response—where the graph and the query sub-tasks *co-evolve* at runtime, yielding dynamic, query-adaptive context that converges to a minimal, sufficient subgraph.

## 2 RELATED WORK

### 2.1 GRAPH-BASED RETRIEVAL-AUGMENTED GENERATION

Recent advances in retrieval-augmented generation (RAG) have increasingly emphasized structural awareness to improve reasoning depth and contextual coherence. Edge et al. (2024) propose GraphRAG, which builds a knowledge graph (KG) from documents via LLM-based entity and relation extraction, then applies community detection to generate hierarchical summaries for global sensemaking. Guo et al. (2024) introduce LightRAG, which employs a dual-level retrieval system combining low-level fact retrieval and high-level semantic discovery using a compact KG, improving both efficiency and coverage. Further building on this idea, Gutiérrez et al. (2024; 2025) present a non-parametric continual learning framework that uses Personalized PageRank over an open KG to enable associative, multi-hop reasoning. Other structure-augmented RAG methods include RAP-TOR (Sarthi et al., 2024), Chen et al. (2023) enhance sense-making but often introduce noise through uncontrolled summarization or lack explicit support for multi-hop reasoning.

### 2.2 KNOWLEDGE GRAPHS IN RAG AND HYBRID APPROACHES

The integration of structured knowledge into LLM reasoning has long been pursued to improve faithfulness and interpretability. Early KG-augmented RAG systems retrieve triples from static external knowledge bases such as Wikidata or Freebase to ground model outputs (Sun et al., 2023). However, these sources are often incomplete, outdated, or misaligned with domain-specific content. To overcome this, hybrid RAG frameworks (Ma et al., 2024) combine unstructured text and structured KGs to balance breadth and precision. Chain-of-Knowledge (CoK) (Li et al., 2024) retrieves from multiple structured sources including Wikipedia, Wikidata, and Wikitable to ground LLM responses. HybridRAG (Sarmah et al., 2024) fuses vector-based and KG-based retrievers, demonstrating superior reasoning performance compared to either modality alone.

### 2.3 ITERATIVE AND REFLECTIVE REASONING IN LLMS

Enabling LLMs to reason iteratively has been shown to improve accuracy and faithfulness. ITER-RETGEN (Shao et al., 2023) introduces an iterative loop that alternates between retrieval and generation, using generated hypotheses to guide further search. Trivedi et al. (2023) combine Chain-of-Thought (CoT) with retrieval, interleaving reasoning steps with evidence gathering, significantly

improving performance on multi-hop QA. Self-RAG (Asai et al., 2023) equips LLMs with reflection tokens to decide when to retrieve and whether the output is hallucinated. ReAct (Yao et al., 2023a) combines reasoning traces with external actions, enabling task decomposition and environment interaction. Other efforts focus on continual learning for LLMs, where RAG serves as a non-parametric alternative to fine-tuning (Shi et al., 2024). Continual pretraining (Jin et al., 2022) and instruction tuning (Zhang et al., 2023) can update model parameters but suffer from catastrophic forgetting (Huang et al., 2024). Model editing methods (Yao et al., 2023b) offer fine-grained updates but struggle with generalization.

## 3 METHODOLOGY

Think-on-Graph 3.0 (ToG-3) introduces a novel *Multi-Agent Context Evolution and Retrieval (MACER)* framework for open-domain question answering.

### 3.1 PROBLEM FORMULATION

Let $\mathcal{D} = \{d_i\}_{i=1}^N$ be a text corpus. The objective is to answer a user query $q$ with an answer $a^*$ that is both accurate and *faithful* to the source corpus, derived from a *minimal, sufficient subgraph* $\mathcal{G}_q^*$ of a heterogeneous graph $\mathcal{G}$ constructed from $\mathcal{D}$:

$$\mathcal{G}_q^* = \arg\min_{\mathcal{G}' \subseteq \mathcal{G}} |\mathcal{G}'| \quad \text{subject to} \quad \text{Suff}(q, \mathcal{G}') = 1, \tag{1}$$

where $\text{Suff}(\cdot, \cdot) \in \{0, 1\}$ is an function judging the sufficiency of a subgraph for answering the query.

Existing methods face a critical dilemma: **(1)** Systems like ToG-1 or 2 rely on high-quality, pre-constructed KGs, limiting their applicability to private or specialized domains. **(2)** Corpus-based GraphRAG methods (e.g., GraphRAG, LightRAG) build a static graph from $\mathcal{D}$ in one go. Their performance is bottlenecked by the quality of this initial graph, which in turn depends heavily on the capability of the LLM used for information extraction.

### 3.2 HETEROGENEOUS GRAPH INDEX: SCHEMA AND CONSTRUCTION

#### 3.2.1 NODE AND EDGE SCHEMA

The Constructor Agent builds a heterogeneous graph $\mathcal{G} = (\mathcal{V}, \mathcal{E})$ with three node types:

- **Chunks** ($\mathcal{C}$): Sentence-level text passages from the corpus.
- **Triplets** ($\mathcal{T}$): Semantic triples $(s, p, o)$ extracted from chunks, annotated with entity and relation types ($\text{type}_s$, $\text{type}_p$, $\text{type}_o$).
- **Communities** ($\mathcal{M}$): Summaries of entity clusters obtained via Leiden clustering on the entity co-occurrence graph, each condensed into an abstract.

Edges are defined by three type relations:

- **OPENREL**$(s, p, o)$: Connects entities $s$ and $o$ via predicate $p$ extracted by the LLM, forming an open-domain semantic triple.
- **MENTIONEDIN**$(t, c)$: Connects a triplet $t$ to the chunk $c$ from which it was extracted.
- **SUMMARYFOR**$(m, e)$: Connects a community summary node $m$ to an entity $e$ that belongs to that community.

This unified schema allows both fine-grained (chunk/triplet) and high-level (community) information to be retrieved seamlessly within a single vector space, effectively addressing the local/global retrieval dichotomy of prior GraphRAG systems.

#### 3.2.2 OFFLINE INDEX CONSTRUCTION

Algorithm 1 in Appendix. B details the one-time construction of the universal index $\mathcal{G}$. A key design choice is the use of a single frozen encoder $E_\theta$ (e.g., jina-mebedding-v3 (Sturua et al., 2024))

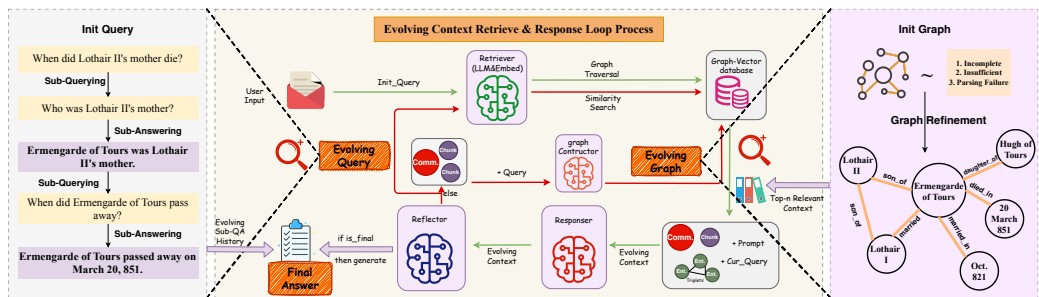

**Figure 3: Multi-Agent Dual-Evolving Context Retrieval-Response Loop.** The Retriever fetches an initial chunk–triplet–community subgraph. The Response Agent produces an answer; the Reflector Agent judges sufficiency (reward=1/0). If insufficient (reward=0), the Reflector evolves the query into sub-queries while the Constructor evolves the subgraph (sub-graph refinement). The loop repeats until the context becomes sufficient or the horizon is reached, after which the Response Agent synthesizes the final answer from the full trajectory.

to embed all nodes—regardless of type—into a unified 1024-dimensional dense vector space. This enables efficient vector search across all node types during retrieval.

## 3.3 THE MACER PROCESS: MULTI-AGENT CONTEXT EVOLUTION AND RETRIEVAL

The core of ToG-3 is the online MACER loop (Algorithm 2), an iterative process of retrieval, generation, and reflection that dynamically evolves the context subgraph $\mathcal{G}_k$. We formalize this process as an episodic Markov Decision Process (MDP) $\mathcal{M} = (\mathcal{S}, \mathcal{A}, P, r)$.

**State Space ($\mathcal{S}$)** : At each step $k$, the state $s_k = (q, \mathcal{G}_k, \mathcal{H}_k)$ captures the complete reasoning context, including the original query $q$, the current evidence subgraph $\mathcal{G}_k$ retrieved by Retriever Agent $\pi_{\text{ret}}$, and the trajectory history $\mathcal{H}_k = (q'_i, a_i, r_i, \mathcal{G}i)_{i=0}^{k-1}$ of all previous sub-queries, answers, rewards, and sub-graphs.

**Action Space ($\mathcal{A}$)** : The Reflector Agent $\pi_{\text{ref}}$ serves as the policy network. Its action $a_k$ at state $s_k$ is either to generate a targeted refinement sub-query $q'_k$ (to continue the reasoning process) or to output the STOP action (to terminate the episode).

**Reward Function ($r$)** : Upon the Response Agent generating an answer $a_k$, the Reflector immediately provides a sparse, binary reward $r_k$:

$$r_k = \begin{cases} 1 & \text{if } \mathtt{Suff}(q, \mathcal{G}_k, a_k) = 1 \quad \text{(sufficient context)} \\ 0 & \text{otherwise.} \end{cases} \tag{2}$$

This reward signal is produced by the Reflector Agent to determine if the current context evidence is sufficient to answer the user's query.

**Transition Dynamics ($P$)** Given the current state $s_k$ and an action $a_k$ (which corresponds to issuing a sub-query $q'k$), the transition to the next state $s_{k+1}$ occurs deterministically according to the following update rules: The constructor agent $\pi_{\text{const}}$ applies the transition operator using the generated sub-query $q'_k$ and the current graph state $\mathcal{G}_k$ to produce an updated graph $\mathcal{G}_{k+1}$. This step including iterative sequence of *evolving queries* and *evolving sub-graphs* reflects the structural evolution of the graph based on the agent's reasoning action, formally defined by the recurrence:

$$q'_k = \pi_{\text{ref}}^{\text{evolve}}(q, \mathcal{G}_k), \tag{3}$$

$$\mathcal{G}_{k+1} = \pi_{\text{const}}^{\text{evolve}}(q'_k, \mathcal{G}_k), \tag{4}$$

The action history $\mathcal{H}_{k+1}$ is augmented with a new tuple recording the executed sub-query $q'_k$, the corresponding action $a_k$, the reward $r_k$ received, and the resulting graph state $\mathcal{G}_{k+1}$. This ensures a comprehensive trace of the reasoning trajectory, which is essential for credit assignment and subsequent learning.

$$\mathcal{H}_{k+1} = \mathcal{H}_k \cup (q'_k, a_k, r_k, \mathcal{G}_{k+1}) \tag{5}$$

$$a^* \leftarrow \pi^{\text{final}}_{\text{resp}}(q, \mathcal{H}_k) \tag{6}$$

The complete MACER process, now cast as an MDP, is summarized in Algorithm 2. The loop continues until the Reflector's policy $\pi_{\text{ref}}$ outputs the STOP action (via $r_k = 1$) or a maximum horizon $K$ is reached. The final answer $a^*$ is synthesized from the full trajectory $\mathcal{H}_k$ of states and actions, ensuring faithfulness to the evolved evidence. This MDP formulation provides the formal foundation for establishing the convergence of the MACER process under mild assumptions, as detailed in Appendix. I. This iterative refinement allows ToG-3 to start from a potentially weak initial graph but *specialize* it towards the reasoning path of the specific query, converging on a high-quality evidence subgraph $\mathcal{G}^*_q$. This evolving and refinement mechanism alleviate the three fundamental weaknesses of small LMs in static GraphRAG, including incomplete triplet recall, insufficient knowledge details and high parsing failure of LLMs' output, as mentioned in Section 1.

## 4 EXPERIMENT

### 4.1 EXPERIMENTAL SETUP

**Datasets**   To comprehensively evaluate the reasoning capabilities of RAG systems, we conduct experiments on two distinct categories of tasks: **Deep Reasoning Tasks** including HotpotQA (Yang et al., 2018), 2WikiMultiHopQA (Ho et al., 2020) and Musique (Trivedi et al., 2022) and **Broad Reasoning Tasks** including 4 subsets of UltraDomain (Qian et al., 2025) benchmark. Detailed statistics for all datasets are provided in Table 4 and Appendix. C.

**Evaluation Metrics**   For **Deep Reasoning Tasks**, we follow standard QA evaluation practices with **Exact Match (EM)** and **F1 Score**. For **Broad Reasoning Tasks**, we adopt a multi-dimensional LLM-based evaluation approach including **Comprehensiveness**, **Diversity** and **Empowerment** following (Guo et al., 2024). Metrics detail are provide Appendix.E.

**Baselines**   We compare ToG-3 against the following state-of-the-art RAG methods across all datasets, including NaiveRAG (Gao et al., 2023), ToG-2 (Ma et al., 2024), GraphRAG (Edge et al., 2024), LightRAG (Guo et al., 2024), MiniRAG (Fan et al., 2025) and HippoRAG-2 (Gutiérrez et al., 2025). Baselines details can be found in Appendix.D. For graph-based methods, we maintain identical chunk sizes (1024 tokens) and use the same LLM (Qwen2.5-32B-Instruct (Yang et al., 2024)) for all extraction and generation tasks to eliminate model capability variations. Implementation details are provide Appendix.A.

### 4.2 RESULT OF DEEP REASONGING BENCHMARK

**Result Analysis from a Method Perspective.**   Results shown in Table 1 represent the average of three independent reasoning experiments. Previsou Graph-based methods like GraphRAG that rely on LLM-based graph construction show limited performance. Their performance is the lowest, particularly in terms of F1 scores as shown in Figure 4b, which can be attributed to a lack of focus on deep factual reasoning and a tendency to produce verbose responses, resulting in low token-level recall. More detailed precision and recall results are provided in Appendix. F.1. ToG-2, without leveraging well-curated knowledge graphs like Freebase and Wikidata, demonstrates moderate performance in open-domain settings. NaiveRAG achieves competitive third-place results by avoiding graph construction limitations and relying solely on retrieved documents for response generation. HippoRAG-2 emerges as the strongest baseline, employing an efficient embedding model with Personalized PageRank algorithm and LLM-based triple filtering to achieve second-best performance. However, our proposed method consistently outperforms all competitors, achieving the highest average EM (0.453) and F1 (0.312) scores across all three benchmarks. This superior performance is attributed to our novel Chunk-Triplets-Community heterogeneous graph architecture and the Multi-Agent Context Evolution and Retrieval (MACER) framework, which enables adaptive subgraph refinement and evolving query decomposition for complex reasoning tasks and overcomes the graph construction challenges that plague other graph-based RAG systems.

**Table 1:** Exact Match (EM) and F1 scores on Deep Reasoning datasets. We highlight the best , second-best , and third-best methods with different background color shades.

| Method | HotpotQA | | 2WikiMultihopQA | | Musique | | Average | |
|---|---|---|---|---|---|---|---|---|
| | EM | F1 | EM | F1 | EM | F1 | EM | F1 |
| NaiveRAG | 0.634 | 0.365 | 0.382 | 0.189 | 0.230 | 0.143 | 0.415 | 0.232 |
| ToG-2 | 0.308 | 0.153 | 0.401 | 0.194 | 0.103 | 0.105 | 0.271 | 0.151 |
| GraphRAG | 0.337 | 0.011 | 0.439 | 0.018 | 0.109 | 0.008 | 0.295 | 0.012 |
| LightRAG | 0.308 | 0.013 | 0.420 | 0.023 | 0.082 | 0.009 | 0.270 | 0.015 |
| MiniRAG | 0.213 | 0.012 | 0.125 | 0.018 | 0.067 | 0.007 | 0.135 | 0.012 |
| HippoRAG-2 | 0.612 | 0.534 | 0.491 | 0.254 | 0.212 | 0.145 | 0.438 | 0.311 |
| Ours | 0.639 | 0.516 | 0.500 | 0.267 | 0.221 | 0.153 | 0.453 | 0.312 |

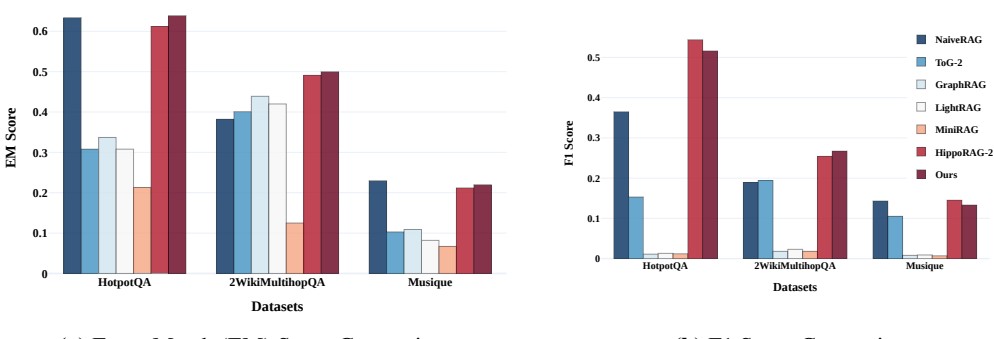

**(a)** Exact Match (EM) Score Comparison      **(b)** F1 Score Comparison

**Figure 4:** Performance comparison of different RAG methods on multi-hop QA datasets. (a) Exact Match scores measure the percentage of questions where the model's answer exactly matches the ground truth. (b) F1 scores provide a harmonic mean of precision and recall for token-level answer matching.

**Result Analysis from a Dataset Perspective.** As shown in Figure 4, the average performance of the baselines and our method across the HotpotQA, 2WikiMultiHopQA, and Musique datasets generally follows a descending trend. This pattern can be attributed to the following reasons: HotpotQA (Yang et al., 2018): Although widely used, this dataset has been shown to provide a weaker test of multi-hop reasoning due to the presence of numerous spurious cues and shortcut signals (Trivedi et al., 2022; Gutiérrez et al., 2024). Musique (Trivedi et al., 2022): A challenging multi-hop QA dataset comprising approximately requiring 2–4 hops, which emphasizes a comprehensive evaluation of multi-step reasoning abilities. Musique is designed to feature diverse and complex reasoning paths, necessitating the integration of information across multiple hops to arrive at correct answers.

### 4.3 RESULT OF BROAD REASONING TASKS

As shwon in Figure 5, The four heatmaps clearly demonstrate that the five methods can be distinctly divided into two clusters: the upper-right region (predominantly red, indicating superior performance) and the lower-left region (predominantly blue, indicating inferior performance). Specifically, ToG-3, GraphRAG, and LightRAG exhibit significantly higher win rates compared to NaiveRAG and HippoRAG-2. Detailed win rates (%) of baselines v.s. ToG-3 across four datasets are provided in Table 6 of Appendix. F. Our framework outperforms NaiveRAG by substantial margins (up to 88.8% overall win rate on Legal dataset and 72.9% average win rate on all four datasets), highlighting the limitations of chunk-based retrieval for complex queries. While GraphRAG shows competitive performance in comprehensiveness due to its extensive community summarization and retrieval, ToG-3 achieves better balance across all metrics, particularly excelling in diversity and empowerment through its heterogeneous graph architecture that integrates chunk-level, triplet-level,

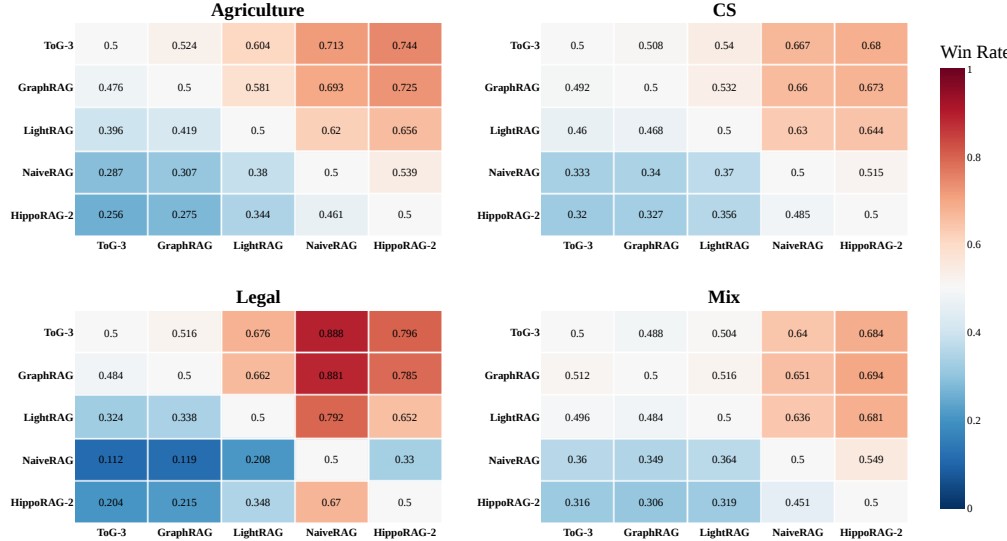

**Figure 5: ELO-based Pairwise Win Rate Matrices Across Four Benchmark Datasets.** Each heatmap visualizes win probabilities derived from direct head-to-head experimental comparisons, transformed through the ELO framework to ensure transitive consistency. The diagonal of the heatmap is set to a default value of 0.5, indicating self-comparison of the method.

**Table 2:** Computational cost comparison across datasets between Graph-based methods. The best EM score of each dataset are marked in **bold**. ToG-3 achieves the best accuracy with efficient indexing and justified inference cost.

| Dataset | Method | Graph Statistics | | | Indexing | Inference | Avg. |
| | | Entities | Relations | Communities | Time (h) | Time (s/q) | EM |
|---|---|---|---|---|---|---|---|
| HotpotQA | ToG-3 | 37,358 | 30,987 | 5,041 | 12.5 | 16.82 | **0.639** |
| | GraphRAG | 94,376 | 73,265 | 10,981 | 15.8 | 8.91 | 0.337 |
| | LightRAG | 94,578 | 76,157 | - | 12.1 | 6.54 | 0.308 |
| 2WikiMultihopQA | ToG-3 | 19,311 | 21,077 | 3,417 | 8.2 | 14.95 | **0.500** |
| | GraphRAG | 50,556 | 37,840 | 6,261 | 10.3 | 7.45 | 0.439 |
| | LightRAG | 50,177 | 37,995 | - | 7.8 | 5.23 | 0.420 |
| Musique | ToG-3 | 32,842 | 39,134 | 6,258 | 9.7 | 13.24 | **0.221** |
| | GraphRAG | 106,042 | 83,139 | 9,407 | 13.2 | 9.37 | 0.109 |
| | LightRAG | 94,621 | 75,923 | - | 10.3 | 7.12 | 0.082 |
| Average | ToG-3 | 29,837 | 30,399 | 4,905 | 10.13 | 15.00 | **0.453** |
| | GraphRAG | 83,658 | 64,748 | 8,883 | 13.10 | 8.58 | 0.295 |
| | LightRAG | 79,792 | 63,358 | - | 10.06 | 6.30 | 0.270 |

and community-level information. Detailed ELO rating calculation for broad reasoning tasks can be found in Appendix. F.3. The multi-agent dual-evolving context retrieval mechanism enables both deep knowledge reasoning through entity-relation exploration and broad community reasoning. This balanced architectural approach makes ToG-3 particularly effective for real-world applications requiring both comprehensive coverage and precise, actionable insights. Our analysis reveals that, on average, 20% of the samples require one evolving-context iteration, 32% require two iterations, and 48% require three iterations. Case studies of ToG-3 retrieval and response output are provided in Appendix. G.

## 4.4 ANALYSIS OF COMPUTATION COST

The Table 2 reveal a consistent accuracy-efficiency trade-off across all datasets. We observed that during the indexing phase, GraphRAG required the longest processing time, averaging 13.10 hours. This is primarily due to its need to extract a large number of triplets and generate community summaries. In comparison, both ToG-3 and LightRAG showed similar indexing times, at 10.13 and 10.06 hours respectively. Although ToG-3 also involves community summary generation, it

**Table 3:** Ablation studies of MACER components and foundation model scaling. Standard ToG-3 settings incorporates all MACER components, employs the Qwen2.5-32B-instruct as the backbone LLM, and utilizes the Jina-v3-embedding model for representation encoding.

| Ablation Setting | HotpotQA | | 2WikiMultihopQA | | Musique | | Average | |
|---|---|---|---|---|---|---|---|---|
| | EM | F1 | EM | F1 | EM | F1 | EM | F1 |
| Standard ToG-3 settings | 0.639 | 0.516 | 0.500 | 0.267 | 0.221 | 0.153 | 0.453 | 0.312 |
| **MACER Components Ablation** | | | | | | | | |
| w/o Evolving Query | 0.598 | 0.443 | 0.412 | 0.203 | 0.178 | 0.121 | 0.396 | 0.256 |
| w/o Evolving Sub-Graph | 0.613 | 0.472 | 0.458 | 0.234 | 0.203 | 0.138 | 0.425 | 0.281 |
| w/o Community Node | 0.641 | 0.519 | 0.487 | 0.259 | 0.216 | 0.148 | 0.448 | 0.309 |
| **Foundation Model Scaling Abalation** | | | | | | | | |
| **LLM Model** | | | | | | | | |
| Qwen2.5-14B | 0.573 | 0.469 | 0.453 | 0.231 | 0.198 | 0.134 | 0.408 | 0.278 |
| Qwen2.5-72B | 0.668 | 0.538 | 0.523 | 0.281 | 0.235 | 0.162 | 0.475 | 0.327 |
| **Embedding Model** | | | | | | | | |
| Qwen3-Embed-0.6B | 0.638 | 0.517 | 0.505 | 0.269 | 0.224 | 0.155 | 0.456 | 0.314 |
| Qwen3-Embed-4B | 0.643 | 0.523 | 0.508 | 0.271 | 0.227 | 0.158 | 0.459 | 0.317 |

constructs the graph more efficiently by extracting fewer relational structures during graph initialization compared to both LightRAG and GraphRAG. While LightRAG achieve faster inference times, they suffer from lower accuracy due to redundant graph elements or simpler retrieval mechanisms. GraphRAG's expensive two-stage indexing yields suboptimal results despite longer processing times. ToG-3 demonstrates an effective balance: its efficient heterogeneous graph construction produces refined knowledge bases across all datasets, and while its multi-agent reasoning requires higher inference time, this cost is directly justified by its best performance on all benchmarks, making it ideal for quality-sensitive applications requiring reliable reasoning capabilities. Note that HippoRAG-2 was excluded from the comparative analysis due to its reliance on OpenIE-based extraction rather than pure LLM extraction, which differs fundamentally from the approaches under investigation.

## 4.5 ABALATION STUDY

**Abalation Study of MACER component** Our ablation study reveals the relative importance of each MACER component for deep reasoning performance. The most significant performance degradation occurs when removing the evolving query mechanism (average performance drop of 12.6% in EM and 17.9% in F1), underscoring its critical role in complex question answering, expecially when using smaller LLMs. The iterative query decomposition enables the framework to break down multifaceted questions into tractable sub-problems, which is essential for navigating the heterogeneous graph structure. Removing subgraph refinement causes a moderate performance decrease (average drop of 6.2% in EM and 10.0% in F1), indicating its importance in adapting the knowledge structure to the specific reasoning context. Interestingly, community nodes show the smallest impact on deep reasoning tasks (a slight drop in the average EM and F1 scores), suggesting that while they contribute to performance, the chunk and triplet representations carry most of the relevant information for precise answer generation. However, in broad reasoning tasks, community nodes are essential for comprehensive coverage and diversity, highlighting the complementary roles of different node types in our heterogeneous graph architecture.

**Abalation Study of used foundation model** The foundation model scaling analysis reveals several important patterns. First, LLM capacity has a substantially greater impact on performance than embedding model size. Scaling from Qwen2.5-14B to Qwen2.5-72B yields a 16.4% average improvement in EM scores, highlighting the critical role of reasoning capability in complex QA tasks. Second, larger embedding models provide consistent but more modest improvements. Qwen3-Embed-0.6B shows a slight average EM improvement over jina-embeddings-v3, while Qwen3-Embed-4B provides a 1.6% improvement. This suggests that while retrieval quality matters and larger embedding models contribute to better performance, the LLM's reasoning capacity remains

the primary bottleneck for complex reasoning tasks. These findings provide practical guidance for resource allocation in real-world deployments.

## 5 CONCLUSION

In this work, we introduced Think-on-Graph 3.0, a novel framework that fundamentally rethinks the paradigm of RAG for complex reasoning. By proposing the Multi-Agent Context Evolution Retrieval (MACER) mechanism and a dynamic Chunk-Triplets-Community heterogeneous graph architecture, we address critical limitations in both existing graph-based RAG methods and knowledge-graph-dependent approaches. Our comprehensive experimental evaluation demonstrates that ToG-3 achieves state-of-the-art performance across multiple challenging benchmarks. The framework's core innovation is its dual-evolving mechanism—comprising Evolving Query and Evolving Subgraph—which dynamically refines both the query representation and the underlying graph structure throughout the reasoning process. This iterative co-evolution enables deep, multi-hop inference while preserving broad coverage across complex queries. This adaptive capability proves particularly valuable for overcoming the quality constraints of static graph construction and the domain limitations of pre-existing knowledge bases. The framework's ability to work with light LLMs also opens possibilities for more efficient and deployable AI systems. We believe the principles established in ToG-3—dynamic graph evolution, multi-agent collaboration, and heterogeneous knowledge integration—provide a foundation for the next generation of RAG systems.

## 6 LIMITATION AND FUTURE DIRECTIONS

Of course our work has several limitations. First, constrained by GPU resources, our experiments are primarily conducted with LLMs up to 72B parameters and embedding models up to 4B parameters—though these sizes are practical for most developers and small-to-medium enterprises for local deployment. Second, the evolving query and sub-graph refinement components increase inference latency, typically 2–3× slower than baseline methods, making our approach more suitable for accuracy-critical applications where sacrificing speed for improved knowledge fidelity is acceptable. Third, the same mechanisms result in longer context inputs, which demand larger GPU memory capacity for efficient processing. These limitations could be mitigated through model distillation, optimized graph traversal algorithms, and dynamic context pruning techniques in future improvement.

Future work will explore three promising directions for further advancement. First, we plan to extend our multi-agent evolving framework to support larger-scale and more complex knowledge-intensive tasks, such as programming assistance, financial analysis, and clinical decision-making. Second, we aim to generalize our method from text to multimodal reasoning, integrating audio, image, and video modalities to construct a world model that bridges textual knowledge with perceptual grounding. Third, inspired by human cognitive science and brain science, we will explore novel architectures that combine parametric memory with large language models to unify memory and reasoning in a seamless framework, enabling more efficient knowledge retention and tool-use capabilities akin to human intelligence.

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

APPENDICES

Within this supplementary material, we elaborate on the following aspects:

- Appendix A: Implementation Details and Hyperparameters
- Appendix B: Detailed ToG-3 Algorithms
- Appendix C: Datasets Statistics and Details
- Appendix D: Baselines Details
- Appendix E: Evaluation Metrics
- Appendix F: More Experiment Results and Details
- Appendix G: Case Study for ToG-3
- Appendix H: Graph Visualization Examples
- Appendix I: Theoretical Support for ToG-3
- Appendix J: LLM Prompts

## A  IMPLEMENTATION DETAILS

We implement ToG-3 experiments with the following configuration: **Data Processing**: Chunk size is set to 1024 tokens with 20-token overlap between consecutive chunks to maintain contextual continuity. **Multi-Agent hyperparameter**: Constructor Agent extracts a maximum of 2 knowledge triplets per chunk and employs hierarchical Leiden clustering (Traag et al., 2019) with maximum cluster size of 5 for community detection. Retriever Agent retrieves top-5 most relevant nodes using hybrid vector-graph similarity matching. Reflector/Responser Agent utilizes the top-5 retrieved passages as context for answer generation. **Backend Infrastructure**: LLM service is based on Qwen2.5-32B-Instruct (Yang et al., 2024) deployed with vLLM (Kwon et al., 2023) engine using bfloat16 precision and prefix caching enabled and greedy-search generation method, which is more stable than the Qwen3 model in mixed reasoning mode in our task; embeddings are generated using Jina-embeddings-v3 (1024-dimensional) (Sturua et al., 2024); Our server is equipped with 8 A100 40GB cards, AMD EPYC 256-core Processor, 2TB memory, and Ubuntu 20.04.1 system. and the hybrid vector-graph storage is implemented using Neo4j community edition [1] for efficient knowledge representation and retrieval, see Appendix.H for visualized graph example.

## B  ToG-3 ALGORITHMS

Algorithms 1 and 2 present the two-stage pipeline of ToG-3. The first stage constructs a heterogeneous graph index comprising chunks, triplets, and communities, while the second stage implements a Multi-Agent Context Evolution and Retrieval (MACER) loop featuring a novel dual-evolution mechanism—Evolving Query and Evolving Subgraph—that dynamically refines both the query representation and the graph structure through iterative interaction.

## C  DATASET DETAIL

This section presents a comprehensive statistical overview of the **Deep and Broad datasets** we use in this paper, including detailed statistics metadata and licensing information, as summarized in Table 4. Additionally, we provide individual descriptions of each dataset to elucidate their respective characteristics and intended use cases.

### C.1  DEEP REASONING DATASETS

- **HotpotQA** (Yang et al., 2018): A crowdsourced question answering dataset built on English Wikipedia, comprising approximately 113K questions. Each question is constructed to require the combination of information from the introductory sections of two Wikipedia

---

[1] https://neo4j.com/product/community-edition

---

**Algorithm 1** Offline Construction of Heterogeneous Index Graph $\mathcal{G}$

---

**Require:** Corpus $\mathcal{D} = \{d_i\}_{i=1}^N$, lightweight LM $\mathcal{L}_{\text{light}}$, encoder $E_\theta$
**Ensure:** Heterogeneous graph $\mathcal{G} = (\mathcal{V}, \mathcal{E})$
1: $\mathcal{V} \leftarrow \emptyset, \mathcal{E} \leftarrow \emptyset$
2: $\mathcal{C} \leftarrow \texttt{SplitIntoChunks}(\mathcal{D})$          $\triangleright$ Sentence-level segmentation
3: $\mathcal{V} \leftarrow \mathcal{V} \cup \mathcal{C}$
4: **for** each chunk $c \in \mathcal{C}$ **do**
5:   $\mathcal{T}_c \leftarrow \mathcal{L}_{\text{light}}(c)$      $\triangleright$ Extract semantic triplets $(s, p, o, \texttt{type}_s, \texttt{type}_p, \texttt{type}_o)$
6:   $\mathcal{V} \leftarrow \mathcal{V} \cup \mathcal{T}_c$
7:   **for** each triplet $t \in \mathcal{T}_c$ **do**
8:    $\mathcal{E} \leftarrow \mathcal{E} \cup \{\textsc{MentionedIn}(t, c)\}$
9:   **end for**
10: **end for**
11: $G_e \leftarrow \texttt{BuildEntityCoOccurrenceGraph}(\mathcal{T})$        $\triangleright$ $\mathcal{T}$ is all triplets
12: $\{M_\ell\}_\ell \leftarrow \texttt{LeidenClustering}(G_e)$
13: **for** each community $M_\ell$ **do**
14:   $m_\ell \leftarrow \mathcal{L}_{\text{light}}(M_\ell)$         $\triangleright$ Generate community summary
15:   $\mathcal{V} \leftarrow \mathcal{V} \cup \{m_\ell\}$
16:   **for** each entity $e \in M_\ell$ **do**
17:    $\mathcal{E} \leftarrow \mathcal{E} \cup \{\textsc{SummaryFor}(m_\ell, e)\}$
18:   **end for**
19: **end for**
20: **Encode** every node $v \in \mathcal{V}$ using $E_\theta$         $\triangleright$ Unified dense encoding
21: **return** $\mathcal{G} = (\mathcal{V}, \mathcal{E})$

---

**Algorithm 2** ToG-3: Multi-Agent Context Evolution and Retrieval (MACER) Loop

---

**Require:** Query $q$, heterogeneous graph $\mathcal{G}$, LLM $\mathcal{L}$, max rounds $K$
**Ensure:** Final answer $a^*$
1: $k \leftarrow 0, \mathcal{G}_0 \leftarrow \texttt{Retriever}(q, \mathcal{G})$        $\triangleright$ Initial retrieval
2: $\mathcal{H}_0 \leftarrow \{(q, \mathcal{G}_0, \text{init})\}$        $\triangleright$ Initialize trajectory history
3: **repeat**
4:   $a_k \leftarrow \pi_{\text{resp}}(q, \mathcal{G}_k, \mathcal{H}_k)$       $\triangleright$ Response Agent generates answer
5:   $r_k \leftarrow \pi_{\text{ref}}^{\text{suff}}(q, \mathcal{G}_k, a_k)$        $\triangleright$ Reflector judges sufficiency
6:   **if** $r_k = 1$ **then break**
7:   **end if**
8:   $q_k' \leftarrow \pi_{\text{ref}}^{\text{evolve}}(q, \mathcal{G}_k)$        $\triangleright$ Reflector evolves query
9:   $\mathcal{G}_{k+1} \leftarrow \pi_{\text{const}}^{\text{evolve}}(q_k', \mathcal{G}_k)$       $\triangleright$ Constructor evolves subgraph
10:   $\mathcal{H}_{k+1} \leftarrow \mathcal{H}_k \cup \{(q_k', a_k, r_k, \mathcal{G}_{k+1})\}$
11:   $k \leftarrow k + 1$
12: **until** $k = K$
13: $a^* \leftarrow \pi_{\text{resp}}^{\text{final}}(q, \mathcal{H}_k)$       $\triangleright$ Synthesize answer from full trajectory
14: **return** $a^*$

---

articles for answering. The dataset provides two gold paragraphs per question, along with a list of sentences identified as supporting facts necessary to answer the question. HotpotQA includes various reasoning strategies such as bridge questions (involving missing entities), intersection questions (e.g., "what satisfies both property A and property B?"), and comparison questions (comparing two entities through a common attribute). It is available in two settings: a *few-shot distractor setting* where models are provided with 10 paragraphs including the gold ones, and an *open-domain full-wiki setting* where models must retrieve relevant passages from the entire Wikipedia corpus given only the question.

- **2WikiMultihopQA** (Ho et al., 2020): A multi-hop question answering dataset that contains complex questions requiring reasoning over multiple Wikipedia paragraphs. Each question is designed to necessitate logical connections across different pieces of information to arrive at the correct answer.

**Table 4:** Statistics of Deep Reasoning and Broad Reasoning Datasets. Metrics abbreviations: Comp. (Comprehensiveness), Div. (Diversity), Emp. (Empowerment).

| Dataset | Corpus Size | Chunks | Entities/Relations | Communities | Metrics | License |
|---|---|---|---|---|---|---|
| **Deep Reasoning Tasks** | | | | | | |
| HotpotQA | 9,809 | 9,812 | 37,358/30,987 | 5,041 | EM, F1 | Apache-2.0 |
| 2WikiMultihopQA | 6,119 | 6122 | 19,311/21,077 | 3,417 | EM, F1 | Apache-2.0 |
| Musique | 11,254 | 11,300 | 32,842/39,134 | 6,258 | EM, F1 | CC-BY-4.0 |
| **Broad Reasoning Tasks** | | | | | | |
| CS | 10 | 2,134 | 3,530/33,507 | 1,166 | | |
| Agriculture | 12 | 2,025 | 6,043/12,571 | 1,039 | Comp., Div., Emp. | Apache-2.0 |
| Legal | 94 | 5,900 | 26,180/44,334 | 1,359 | | |
| Mix | 61 | 658 | 2,784/5,089 | 425 | | |

- **Musique** (Trivedi et al., 2022): A challenging multi-hop QA dataset containing approximately 25K 2–4 hop questions, constructed by composing single-hop questions from five existing single-hop QA datasets. It is designed to feature diverse and complex reasoning paths, requiring models to integrate information from multiple hops to generate correct answers. The dataset emphasizes comprehensive evaluation of multi-step reasoning capabilities.

## C.2 BROAD REASONING DATASETS

The following datasets are curated from the UltraDomain (Qian et al., 2025) benchmark. The benchmark construction leverages financial reports, legal contracts, and 428 college textbooks across 18 distinct domains to evaluate model versatility and adaptability in specialized and broad application scenarios:

- **CS**: Computer science domain focusing on data science, software engineering, and programming topics, requiring technical comprehension and analytical reasoning.

- **Agriculture**: Covers agricultural practices including beekeeping, crop production, and disease prevention, demanding domain-specific knowledge integration.

- **Legal**: Derived from legal contracts and documents, focusing on corporate legal practices, regulatory compliance, and governance, requiring precise interpretation of nuanced legal language.

- **Mix**: Contains diverse contexts from college textbooks spanning natural sciences, humanities, and social sciences, testing generalization capabilities across interdisciplinary topics.

## D BASELINES

This section presents the baseline methods evaluated in this paper, encompassing both classical algorithms such as NaiveRAG and GraphRAG, as well as recently proposed approaches including LightRAG, ToG-2, and HippoRAG-2. Baselines are as follows:

- **NaiveRAG** (Gao et al., 2023): A standard chunk-based retrieval baseline that segments raw texts into chunks and stores them in a vector database using text embeddings. For queries, it generates vectorized representations to directly retrieve text chunks based on semantic similarity.

- **GraphRAG** (Edge et al., 2024): A graph-enhanced RAG system that utilizes an LLM to extract entities and relationships from text, representing them as nodes and edges. It generates community summaries through graph clustering and employs both local (entity-based) and global (community-based) retrieval strategies for comprehensive information access.

- **LightRAG** (Guo et al., 2024): A graph-structured RAG framework that employs a dual-level retrieval system combining low-level entity retrieval with high-level knowledge discovery. It integrates graph structures with vector representations for efficient retrieval of related entities and their relationships.

- **ToG-2** (Ma et al., 2024): A knowledge graph-based framework implements a tight-coupling hybrid RAG paradigm that iteratively retrieves information from both unstructured texts and structured knowledge sources. It alternates between graph retrieval and context retrieval for in-depth knowledge exploration.
- **HippoRAG-2** (Gutiérrez et al., 2025): A non-parametric continual learning framework that leverages Personalized PageRank algorithm over an open knowledge graph constructed using LLM-extracted triples. It enhances multi-hop reasoning capabilities through sophisticated graph traversal and passage integration mechanisms.

## E   METRICS

We employ different evaluation protocols for the two task categories:

For **Deep Reasoning Tasks**, we follow standard QA evaluation practices as ToG (Sun et al., 2023; Ma et al., 2024) and HippoRAG (Gutiérrez et al., 2024; 2025):

- **Exact Match (EM)**: Measures the percentage of predictions that exactly match the ground truth answer. specifically, we use Sub-string Exact Match version (Sun et al., 2023; Ma et al., 2024) which goes through the whole response to check whether the answer is in.
- **F1 Score**: Computes word-level overlap between predictions and ground truth answers.

For **Broad Reasoning Tasks**, we adopt a multi-dimensional LLM-based evaluation approach due to the complexity and open-ended nature of these queries following LightRAG (Guo et al., 2024):

- **Comprehensiveness (Comp.)**: Measures how thoroughly the answer addresses all aspects of the question.
- **Diversity (Div.)**: Assesses the variety of perspectives and insights provided in the answer.
- **Empowerment (Emp.)**: Evaluates how well the answer enables informed understanding and judgment.

The LLM-based evaluation uses GPT-4o-mini as judge, with careful attention to prompt design and answer ordering to avoid positional bias. The LLM evaluation prompt is shown in Appendix.J

## F   MORE EXPERIMENT RESULTS AND DETAILS

This section presents extended experimental results, including detailed precision and recall metrics on Deep Reasoning tasks, as well as one-to-one win rates from Broad Reasoning tasks. The pairwise win rates are converted into a unified ELO rating system, with the resulting ratings visualized in the heatmap shown in Figure 5.

### F.1   PRECISION AND RECALL RATE RESULTS

Table 5 reveals the underlying reason for the relatively low F1 scores of GraphRAG and LightRAG: these methods are not specifically designed for deep reasoning tasks. By examining both precision/recall metrics and output cases, we observe that excessively long or unfocused responses tend to substantially reduce recall, thereby diminishing overall F1 performance.

### F.2   RESULT DETAIL IN BRAOD REASONING TASKS

Table 6 presents the pairwise win rates (%) of baseline methods against ToG-3 across four datasets and four evaluation dimensions. The results demonstrate that ToG-3 consistently outperforms all compared baselines.

### F.3   ELO RATING CALCULATION FOR BROAD REASONING TASKS

This appendix details the mathematical framework and computational process for deriving ELO ratings from pairwise comparison data across four benchmark datasets. The ELO rating system provides a mathematically consistent approach to quantify relative performance differences between

**Table 5:** Comprehensive Evaluation Metrics of five RAG methods across three deep reasoning datasets. The best results of each dataset are marked in **bold**.

| Method | HotpotQA | | | 2WikiMultihopQA | | | Musique | | |
|---|---|---|---|---|---|---|---|---|---|
| | F1 | R | P | F1 | R | P | F1 | R | P |
| NaiveRAG | 0.365 | 0.593 | 0.346 | 0.189 | 0.345 | 0.168 | 0.143 | 0.280 | 0.126 |
| GraphRAG | 0.011 | 0.423 | 0.006 | 0.018 | 0.456 | 0.009 | 0.008 | 0.266 | 0.004 |
| LightRAG | 0.013 | 0.393 | 0.007 | 0.023 | 0.429 | 0.012 | 0.009 | 0.224 | 0.005 |
| MiniRAG | 0.012 | 0.372 | 0.006 | 0.018 | 0.403 | 0.009 | 0.007 | 0.203 | 0.003 |
| ToG-3 | **0.516** | **0.595** | **0.454** | **0.267** | **0.485** | **0.312** | **0.153** | **0.286** | **0.132** |

P: Precision, R: Recall. ToG-3 achieves best F1 while maintaining high precision-recall balance.

**Table 6:** Win rates (%) of baselines v.s. ToG-3 across four datasets and four evaluation dimensions. The better results of each dataset are marked in **bold**.

| Metrics | Agriculture | | CS | | Legal | | Mix | |
|---|---|---|---|---|---|---|---|---|
| | NaiveRAG | ToG-3 | NaiveRAG | ToG-3 | NaiveRAG | ToG-3 | NaiveRAG | ToG-3 |
| Comprehensiveness | 28.4% | **71.6%** | 32.4% | **67.6%** | 12.4% | **87.6%** | 34.8% | **65.2%** |
| Diversity | 19.6% | **80.4%** | 32.0% | **68.0%** | 9.6% | **90.4%** | 28.4% | **71.6%** |
| Empowerment | 29.4% | **70.6%** | 32.8% | **67.2%** | 12.4% | **87.6%** | 38.8% | **61.2%** |
| Overall | 28.7% | **71.3%** | 33.3% | **66.7%** | 11.2% | **88.8%** | 36.0% | **64.0%** |
| | GraphRAG | ToG-3 | GraphRAG | ToG-3 | GraphRAG | ToG-3 | GraphRAG | ToG-3 |
| Comprehensiveness | 46.8% | **53.2%** | 49.6% | **50.4%** | 49.6% | **50.4%** | **51.6%** | 48.4% |
| Diversity | 44.4% | **55.6%** | 48.4% | **51.6%** | 46.8% | **53.2%** | **52.0%** | 48.0% |
| Empowerment | 25.2% | **74.8%** | 43.2% | **56.8%** | 29.6% | **70.4%** | 38.4% | **61.6%** |
| Overall | 47.6% | **52.4%** | 49.2% | **50.8%** | 48.4% | **51.6%** | **51.2%** | 48.8% |
| | LightRAG | ToG-3 | LightRAG | ToG-3 | LightRAG | ToG-3 | LightRAG | ToG-3 |
| Comprehensiveness | 38.9% | **61.1%** | 45.6% | **54.4%** | 33.6% | **66.4%** | 47.6% | **52.4%** |
| Diversity | 32.0% | **68.0%** | 42.0% | **58.0%** | 28.0% | **72.0%** | 39.2% | **60.8%** |
| Empowerment | 40.5% | **59.5%** | 46.0% | **54.0%** | 33.6% | **66.4%** | 52.0% | **48.0%** |
| Overall | 39.6% | **60.4%** | 46.0% | **54.0%** | 32.4% | **67.6%** | 49.6% | **50.4%** |
| | HippoRAG-2 | ToG-3 | HippoRAG-2 | ToG-3 | HippoRAG-2 | ToG-3 | HippoRAG-2 | ToG-3 |
| Comprehensiveness | 24.5% | **75.5%** | 31.6% | **68.4%** | 21.6% | **78.4%** | 29.6% | **70.4%** |
| Diversity | 18.8% | **81.2%** | 28.0% | **72.0%** | 17.2% | **82.8%** | 23.6% | **76.4%** |
| Empowerment | 27.8% | **72.2%** | 32.9% | **67.1%** | 21.6% | **78.4%** | 34.0% | **66.0%** |
| Overall | 25.6% | **74.4%** | 32.0% | **68.0%** | 20.4% | **79.6%** | 31.6% | **68.4%** |

retrieval-augmented generation methods. The ELO rating system transforms raw win rates into a logarithmic scale that ensures transitive consistency in performance rankings. The core transformation is defined as follows:

For a given method $i$ with win rate $w_i$ against the reference method (ToG-3), the ELO rating difference is calculated as:

$$\Delta R_i = 400 \cdot \log_{10}\left(\frac{1}{w_i} - 1\right)$$

The absolute ELO rating for method $i$ is then:

$$R_i = R_{\text{ref}} - \Delta R_i$$

where $R_{\text{ref}} = 1600$ is the reference rating for ToG-3.

The win probability between any two methods $i$ and $j$ with ratings $R_i$ and $R_j$ is given by:

$$P(i \text{ beats } j) = \frac{1}{1 + 10^{(R_j - R_i)/400}}$$

# G  CASE STUDY FOR TOG-3

This section provides a detailed case study of ToG-3 in deep reasoning task (Figure 6) and broad reasoning task (Figure 7 and Figure 8), offering an intuitive demonstration of the execution dynam-

```
Question:
What nationality is the performer of the song When The Stars Go Blue?

Initial Evidence (Sub-Graph):
When The Stars Go Blue -> performed_by -> Ryan Adams
Ryan Adams -> known_for -> alternative country, rock, folk
Ryan Adams -> also_includes_genre -> indie rock, Americana
... (no nationality information present)

Evolving Query Decomposition 1 (Reflector Agent)
Who is the performer of the song When The Stars Go Blue?
Ryan Adams.

Evolving Sub-Graph Refinement (Constructor Agent):
Sammy Hagar -> is -> American rock vocalist
Chuck Berry -> is -> American singer and songwriter
Elvis Presley -> pioneered -> Rock and Roll (American origin)
Ryan Adams -> nationality -> American   [Added]
Wee Willie Walker -> was -> American soul and blues singer

Evolving Query Decomposition 2 (Reflector Agent)
What is Ryan Adams known for besides the song When The Stars Go Blue?
Ryan Adams is known for his work as a singer-songwriter and musician, with a musical style that
encompasses alternative country, rock, and folk.

Evolving Query Decomposition 3 (Reflector Agent)
What is Ryan Adams's nationality?
Ryan Adams is an American singer and musician.

Final Answer (Response Agent):
ryan adams is american.

Ground Truth:
america

Explanation:
The initial sub-graph contained no explicit information about Ryan Adams' nationality.
Through evolving query decomposition, the system identified a missing link and triggered sub-graph
refinement.
New evidence from biographical triples (e.g., "Ryan Adams -> nationality -> American") was
integrated, enabling a definitive answer.
```

**Figure 6:** Case Study: Evolving Query and Evolving Sub-Graph for Multi-Hop Reasoning. This example demonstrates how an initial knowledge sub-graph fails to answer a complex question, prompting a decomposition into a sub-query. The Constructor Agent refines the sub-graph with additional biographical facts, enabling the Response Agent to generate the correct answer. The process highlights the dynamic, iterative nature of self-evolving context retrieval.

ics of its dual-evolution mechanism—comprising Evolving Query and Evolving Subgraph—across multi-step reasoning processes.

## H  GRAPH VISUALIZATION EXAMPLES

This section details two constructed graph used in our study: the 2WikiMultihopQA subset (exemplifying deep reasoning) and the computer science domain graph from UltraDomain (exemplifying broad reasoning), which are visualized with Neo4j community edition [2].

**2WikiMultihopQA Dataset: Exemplar of Depth Reasoning**  2WikiMultihopQA is designed to test depth reasoning—the ability to perform multi-step, sequential inference over entity-relation paths. Each question requires traversing at least two "hops" (e.g., first identifying a person's profession, then linking that profession to a historical event, and finally combining both to answer a causal query). This structure forces models to engage in complex semantic chaining, where errors in early steps propagate, challenging robustness in long-range dependency handling. The dataset's sparse yet densely connected knowledge graphs emphasize precision in step-by-step reasoning over

---

[2]https://neo4j.com/product/community-edition

```
Question:
How do different regression metrics (e.g., RMSE, MAE, R²) vary by model and dataset?

Evidence (from ToG-3):
Mean Squared Error -> is_related_to -> Root Mean Squared Error...
Mean Squared Error -> is_similar_to -> Mean Absolute Error...
summary_23 -> SUMMARY_FOR -> RMSE.MSE emphasizes larger errors...
summary_287 -> SUMMARY_FOR -> MAE. ARIMA model performance evaluated using MAE...

Answer 1(from ToG-3):
Different regression metrics such as RMSE, MAE, and R² can vary by model and dataset...
RMSE penalizes large errors more heavily due to squared terms, while MAE is robust to outliers...
R² measures explained variance but can be misleading on small or overfitted datasets...

Answer 2(from GraphRAG):
The provided community summary does not detail specific variations...To evaluate how these
metrics vary:
1. Train different regression models on your datasets...
2. Compute RMSE, MAE, R² using MLlib.RegressionMetrics...
3. Compare performance across models and datasets...

Evaluation Result (JSON):

{
  "Comprehensiveness": {
    "Winner": "Answer 1",
    "Explanation": "Answer 1 provides a detailed explanation of various regression metrics..."
  },
  "Diversity": {
    "Winner": "Answer 2",
    "Explanation": "Answer 2 presents a different perspective by incorporating specific tools..."
  },
  "Empowerment": {
    "Winner": "Answer 1",
    "Explanation": "Answer 1 empowers the reader by explaining what each metric means..."
  },
  "Overall Winner": {
    "Winner": "Answer 1",
    "Explanation": "Answer 1 is the overall winner as it provides a comprehensive understanding..."
  }
}
```

**Figure 7:** Case Study: Comparing Regression Metrics Across Models and Datasets. This example illustrates how two reasoning systems answer a technical ML question: GraphRAG emphasizes practical implementation (e.g., using Spark's MLlib), while ToG3 focuses on theoretical distinctions between RMSE, MAE, and $R^2$. An evaluator selects the more comprehensive and empowering answer based on evidence from the knowledge graph.

```
Question:
What policy changes are needed to foster equitable access to nutritious food?

Evidence (from ToG-3):
summary_216 -> SUMMARY_FOR -> Food Security. **Supermarket and Food Security**: The reliance on
supermarkets for food is viewed as a threat...
Victory Garden -> promoted -> Food Security... Supermarket -> threatens -> Food Security..."
White House Childhood Obesity Task Force -> recommended -> Farm to School...
Food Works -> collaborates_with -> Sauvie Island Organics...
Michael Taylor -> interprets -> Food Modernization Act... background in representing Monsanto...

Answer 1(from ToG-3):
policy changes that could foster equitable access to nutritious food may include implementing
subsidies...
increasing funding for farm to school programs in underserved communities...
establishing regulations that prioritize the availability of nutritious options in school meal
programs...
and creating educational initiatives to promote awareness about nutrition and healthy eating
habits...
these changes could help address disparities in access to nutritious food...

Answer 2(from GraphRAG):
1. **strengthen conflict of interest policies**: prevent individuals with significant industry
ties...
2. **enhance transparency and accountability**: implement transparent processes...
3. **support diverse food producers**: promote a variety of food producers...
4. **expand nutrition education and assistance programs**: enhance nutrition education...
5. **ensure regulatory independence**: guarantee that regulatory bodies are independent...
6. **promote local food systems**: encourage local food production through grants, subsidies...
7. **support community gardens and urban agriculture**: provide funding and resources...
8. **promote sustainable farming practices**: offer incentives for sustainable farming...
These measures aim to create a more resilient and sustainable food system...

Evaluation Result (JSON):

{
{
  "Comprehensiveness": {
    "Winner": "Answer 2",
    "Explanation": "Answer 2 provides a more detailed and structured set of policy changes... "
    "broader scope than Answer 1..."
  },
  "Diversity": {
    "Winner": "Answer 2",
    "Explanation": "Answer 2 presents a wider variety of perspectives... "
    "including regulatory, community, and economic solutions."
  },
  "Empowerment": {
    "Winner": "Answer 1",
    "Explanation": "Answer 1 empowers readers by offering a comprehensive overview..."
    "facilitates informed judgment."
  },
  "Overall Winner": {
    "Winner": "Answer 2",
    "Explanation": "Answer 2 emerges as the overall winner due to its superior"
    "comprehensiveness, diversity, and empowerment."
  }
}
```

**Figure 8:** Case Study: Policy Recommendations for Equitable Food Access. This example illustrates the full reasoning pipeline: a complex policy question is answered by two different systems (GraphRAG and ToG-3), supported by retrieved knowledge snippets. An evaluator then compares both responses across multiple dimensions, selecting the more comprehensive, diverse, and empowering answer as the winner.

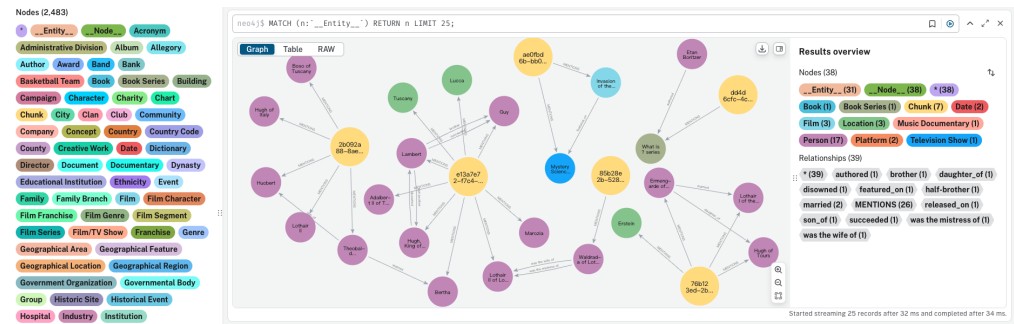

**Figure 9:** Structural overview of the 2WikiMultihopQA subset, exemplifying depth reasoning through multi-hop entity-relation paths (e.g., traversing "person → profession → historical event" to answer causal queries).

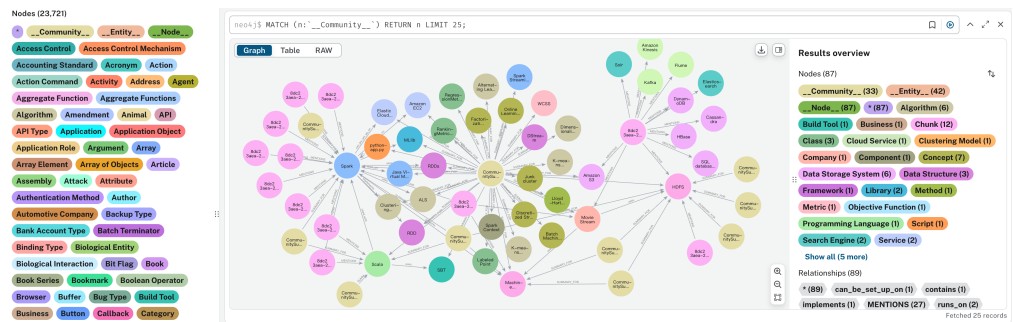

**Figure 10:** Visualization of the computer science domain graph in UltraDomain, showcasing breadth reasoning via diverse node types (e.g., programming languages like Scala/Spark, frameworks like HDFS/Kafka) and relationship types (e.g., implements, runs_on, contains).

surface-level pattern matching. A structural overview highlighting its multi-hop nature is shown in Figure 9.

**Computer Science Domain Graph in UltraDomain: Exemplar of Breadth Reasoning** The computer science domain graph from UltraDomain represents breadth reasoning—focused on expansive coverage of concepts and their interrelations. It includes a wide range of CS entities (from foundational data structures/algorithms to applied distributed systems/cloud services) and diverse relationship types (e.g., implements, runs_on, contains). This breadth challenges models to navigate a large, heterogeneous concept space, where connections span disparate subfields (e.g., linking a programming language to a database, or an algorithm to hardware). For instance, understanding how Spark relates to Hadoop, Kafka, and multiple programming languages requires integrating knowledge across multiple domains, reflecting the need for broad, cross-concept awareness. A visualization of this graph, illustrating its extensive node and edge diversity, is provided in Figure 10.

# I  THEORETICAL SUPPORT: IMPLICIT DYNAMICS OF IN-CONTEXT LEARNING

The iterative refinement process in MACER and dual-evolving mechanism is not merely heuristic but possesses theoretical grounding through the lens of implicit in-context learning dynamics. Recent work by (Dherin et al., 2025) demonstrates that transformer-based models can perform in-context learning by implicitly modifying their MLP weights through attention mechanisms. We extend this theoretical framework to explain the convergence properties of our multi-agent reasoning process.

**Implicit Weight Updates via Attention Dynamics** The trajectory history $\mathcal{H}_k$ serves as an *in-context prompt* that induces implicit low-rank updates to the frozen LLM's parameters. Specifically,

for a transformer module with MLP layer weights $W$, the context $\mathcal{H}_k$ generates an implicit weight update $\Delta W_k$ through the attention mechanism:

$$\Delta W_k = \frac{(W \Delta A_k) A(q)^\top}{\|A(q)\|^2}, \quad \text{where} \quad \Delta A_k = A(\mathcal{H}_k, q) - A(q). \tag{7}$$

Here, $A(\cdot)$ denotes the activation pattern from the attention layer, $A(q)$ represents the baseline activation without context, and $A(\mathcal{H}_k, q)$ captures the contextualized activation with the full reasoning history. The term $\Delta A_k$ quantifies the information injected by the evolving context $\mathcal{H}_k$. The low-rank nature of $\Delta W_k$ ensures efficient and targeted parameter updates without catastrophic forgetting of pre-trained knowledge.

**MDP Policy as an Implicit Function of Context**  Recall from Section 3.3 that the Reflector Agent's policy $\pi_{\text{ref}}$ maps states $s_k = (q, \mathcal{G}_k, \mathcal{H}_k)$ to actions (sub-queries or STOP). Under the implicit learning view, $\pi_{\text{ref}}$ is not a fixed network but an emergent policy $\pi_k$ shaped by $\Delta W_k$. Thus, the sequence $\{\pi_k\}_{k=1}^K$ constitutes a trajectory of implicitly adapted policies driven by the evolving context $\mathcal{H}_k$.

**Convergence via Regret Minimization**  We analyze convergence through the lens of episodic regret minimization in the MDP $\mathcal{M} = (\mathcal{S}, \mathcal{A}, P, r)$. Let $V_{s_k}^\pi = \mathbb{E}_\pi \left[ \sum_{i=k}^K \gamma^{i-k} r_i \mid s_k \right]$ denote the value of policy $\pi$ at state $s_k$, and let $V_{s_k}^* = \max_\pi V_{s_k}^\pi$ be the optimal value. The cumulative regret over $K$ steps is:

$$\mathcal{R}(K) = \sum_{k=1}^K \left( V_{s_k}^* - V_{s_k}^{\pi_k} \right). \tag{8}$$

We establish sublinear regret growth $\mathcal{R}(K) = o(K)$ under the following mild assumptions:

**Assumption 1** (Realizability). *There exists a policy $\pi^*$ such that $\mathsf{Suff}(q, \mathcal{G}_q^*) = 1$, and $\pi^*$ is representable by the implicit policy class induced by in-context prompts of the form $(\mathcal{H}; q)$.*

**Assumption 2** (Bounded Gradient Norm). *The implicit gradient direction $g_k$, defined as the reward-sensitive update signal from $\mathcal{H}_k$, satisfies $\|g_k\| \leq G$ for some constant $G > 0$.*

Under these assumptions, the following properties hold:

**Property 1 (Smooth Policy Evolution).** The value function evolves smoothly with respect to implicit updates:

$$\|V^{\pi_{k+1}} - V^{\pi_k}\|_\infty \leq L\|g_k\| + \mathcal{O}(\|g_k\|^2), \tag{9}$$

for some Lipschitz constant $L > 0$, ensuring stable policy transitions.

**Property 2 (Expected Policy Improvement).** Each refinement step yields non-negative expected improvement:

$$\mathbb{E}\left[ V_{s_k}^{\pi_{k+1}} - V_{s_k}^{\pi_k} \mid \mathcal{H}_k \right] \geq \eta\|g_k\|^2 - \sigma_k, \tag{10}$$

where $\eta > 0$ and $\{\sigma_k\}$ is a martingale difference sequence with $\mathbb{E}[\sigma_k \mid \mathcal{H}_k] = 0$. This follows from the fact that evolving sub-queries generated by the Reflector target knowledge gaps, and the Constructor's evolving graph refinement increases the likelihood of sufficiency.

**Property 3 (Vanishing Implicit Gradient).** As the context becomes increasingly informative, the room for improvement diminishes:

$$\lim_{k \to \infty} \|g_k\| = 0 \quad \text{almost surely.} \tag{11}$$

This is guaranteed by Assumption 1 (Realizability) and the finite horizon $K$, which ensures the process either reaches a sufficient subgraph ($r_k = 1$) or exhausts its budget.

Together, these properties imply that the sequence $\{\pi_k\}$ converges to a policy $\pi^\dagger$ satisfying $V_{s_1}^{\pi^\dagger} \geq V_{s_1}^* - \epsilon$ for arbitrarily small $\epsilon > 0$ as $K \to \infty$. In practice, with a reasonable horizon (e.g., $K = 3$), MACER reliably converges to a sufficient context $\mathcal{G}_q^*$ for faithful answer synthesis.

This analysis establishes that the MACER loop performs an implicit form of policy gradient ascent on the reward landscape defined by context sufficiency, with convergence guarantees rooted in stochastic approximation theory and in-context learning dynamics, providing rigorous foundations for the empirical effectiveness of our reward-based evolving context mechanism.

## J  PROMPT TEMPLATES

Our framework employs a multi-stage, prompt-driven reasoning pipeline that integrates structured knowledge graph (KG) extraction, community-based summarization, iterative sub-query decomposition, sub-graph refinement, and faithful answer synthesis. Each stage is governed by a specialized prompt template designed to ensure modularity, interpretability, and factual consistency. The complete sequence of prompts is as follows:

1. **KG Triplets Extraction**: As shown in Figure 11, given raw textual input, this prompt instructs the model to extract structured subject-relation-object triples (e.g., `entity1 -> relation -> entity2`) to construct a fine-grained knowledge sub-graph. This step transforms unstructured text into a queryable graph structure.

2. **Generate Community Summary**: As shown in Figure 12, based on densely connected sub-graphs (communities), this prompt synthesizes a concise natural language summary that captures the core themes and relationships within each community, enabling high-level semantic indexing and retrieval.

3. **Keyword Expansion for Retrieval Augmentation**: As shown in Figure 13, to improve recall in the querying phase, this prompt generates a set of synonyms and related terms from the original query, considering variations in capitalization, pluralization, and common phrasings, separated by delimiter symbols.

4. **Evolving Sub-Query Decomposition**: As shown in Figure 14, for complex multi-hop questions, this prompt recursively decomposes the current query into simpler, context-answerable sub-questions, guided by previously retrieved information and reasoning traces, enabling stepwise information gathering.

5. **Evolving Sub-Graph Refinement**: As shown in Figure 15, this prompt cleans and enhances the retrieved or extracted sub-graph by removing irrelevant triples, normalizing entity names, and optionally filling in strongly supported missing links, thereby improving the signal-to-noise ratio for downstream reasoning.

6. **Final Answer Synthesis**: As shown in Figure 16, in the final stage, the model generates a concise, context-grounded answer using *only* the refined evidence, with explicit instructions to avoid hallucination or reliance on prior knowledge. If the answer cannot be determined, it returns "Unknown" to maintain factual integrity.

These prompts work in concert to enable structured, interpretable, and reliable reasoning over hybrid text-and-graph knowledge sources. And Figure 17 shows the LLM evaluation prompt in the broad reasoning task. Their modular design allows for independent tuning and auditing, making the overall system transparent and robust to noise and ambiguity.

```
-Goal-
Given a text document, identify all entities and their entity types from the text and all
relationships among the identified entities.
Given the text, extract up to {max_knowledge_triplets} entity-relation triplets.

-Steps-
1. Identify all entities. For each, extract:
entity_name | entity_type | entity_description

2. Identify all related (source, target) pairs. For each, extract:
source_entity | target_entity | relation | relationship_description

3. Output valid JSON only:
{ "entities": [...], "relationships": [...] }

-An Output Example-
{
"entities": [
{ "entity_name": "Albert Einstein", "entity_type": "Person", "entity_description": "..." },
{  "entity_name":   "Theory  of  Relativity",  "entity_type":   "Scientific  Theory",
"entity_description": "..." },
{ "entity_name": "Nobel Prize in Physics", "entity_type": "Award", "entity_description":
"..." }
],
"relationships": [
{ "source_entity": "Albert Einstein", "target_entity": "Theory of Relativity", "relation":
"developed", "relationship_description": "..." },
{ "source_entity": "Albert Einstein", "target_entity": "Nobel Prize in Physics", "relation":
"won", "relationship_description": "..." }
]
}

-Real Data-
####################
text: {text}
####################
output: ;
```

**Figure 11:** KG Triplets Extraction Prompt Template. The template provides structured instructions for extracting entities and relationships from text, with clear formatting for both input requirements and JSON output format.

```
role="system"
You are provided with a set of relationships from a knowledge graph, each represented as
entity1 -> entity2 -> relation -> relationship_description.
Your task is to create a summary of these relationships. The summary should include: Names
of the entities involved, A concise synthesis of the relationship descriptions. The goal is
to capture the most critical and relevant details that highlight the nature
and significance of each relationship. Ensure the summary is coherent and integrates
information to emphasize key aspects. Avoid redundancy and maintain clarity.

role="user"
####################
text: {community_info}
####################

assistant:
% Generated summary based on {community_info} will appear here.
```

**Figure 12:** Community Summary Template. This template provides structured instructions for extracting entities and relationships from text, with clear formatting for input specifications and expected JSON-like output format.

```
role="system"
Given some initial query, generate synonyms or related keywords up to {max_keywords} in
total,
considering possible cases of capitalization, pluralization, common expressions, etc.
Provide all synonyms/keywords separated by '^' symbols: 'keyword1^keyword2^...'.
Note: result should be in one line, separated by '^' symbols.

role="user"
----
QUERY: {query_str}
----

assistant:
% Example: KEYWORDS: machine learning^ML learning machines^AI models^neural networks^deep
learning ...
```

**Figure 13:** Keyword Expansion Prompt Template. This template instructs the model to generate up to {max_keywords} synonyms or related terms for a given query, formatted as a single line separated by '^' symbols.

```
role="system"
The original question is as follows: {query_str}
We have an opportunity to answer some, or all of the question from a knowledge source.
Context information for the knowledge source is provided below, as well as previous reasoning
steps.
Given the context and previous reasoning, return a question that can be answered from the
context.
This question can be the same as the original question, or represent a subcomponent.
It should not be irrelevant to the original question.
If no further information can be extracted, return 'None'.

Examples:
Question: How many Grand Slam titles does the winner of the 2020 Australian Open have?
Knowledge source context: Provides names of the winners of the 2020 Australian Open
Previous reasoning: None
Next question: Who was the winner of the 2020 Australian Open?

Question: How many Grand Slam titles does the winner of the 2020 Australian Open have?
Knowledge source context: Includes biographical info for each winner
Previous reasoning:
- Who was the winner of the 2020 Australian Open?
- The winner was Novak Djokovic.
Next question: How many Grand Slam titles does Novak Djokovic have?

Current Input:
Question: {query_str}
Knowledge source context: {context_str}
Previous reasoning: {prev_reasoning}

assistant:
% Output: <decomposed sub-question> OR 'None'
```

**Figure 14:** Step-wise Query Evolution and Decomposition Prompt Template. This template guides the model to recursively break down a complex question into answerable sub-questions based on available context and prior reasoning, enabling multi-hop reasoning over knowledge sources.

```
role="system"
You are given a sub-graph extracted from a knowledge graph, represented as a list of triples:
entity1 -> relation -> entity2.
This sub-graph may contain irrelevant, redundant, or incomplete information.
Your task is to refine the sub-graph by:
Removing irrelevant or noisy triples not related to the query, Filling in missing but inferable
relationships (if strongly supported),
Ensuring entity names are normalized (e.g., consistent capitalization, singular/plural).
Return the refined sub-graph in the same triple format, one per line.
If no refinement is needed, return the original sub-graph.
If all triples are irrelevant, return 'None'.

Example Input:
Query: What are the major achievements of Marie Curie?
Sub-graph:
Marie Curie -> won -> Nobel Prize in Physics
Marie Curie -> born in -> Warsaw
Marie Curie -> spouse -> Pierre Curie
Apple Inc. -> founded by -> Steve Jobs

Refined Output:
Marie Curie -> won -> Nobel Prize in Physics
Marie Curie -> won -> Nobel Prize in Chemistry
Marie Curie -> spouse -> Pierre Curie
(Note: Added Chemistry prize based on strong prior knowledge; removed birthplace and unrelated
Apple fact)

Current Input:
Query: {query_str}
Sub-graph:
{subgraph_triples}

assistant:
```

**Figure 15:** Sub-Graph Evolution and Refinement Prompt Template. This template guides the model to clean, complete, and normalize a noisy or incomplete knowledge sub-graph in response to a given query, improving its relevance and coherence for downstream reasoning.

```
role="system"
Context information is provided below.
You must answer the query using only this context, and not any prior knowledge.
Do not make assumptions or add information not present in the context.
If the answer cannot be determined from the context, respond with 'Unknown'.

--------------------
{context_str}
--------------------

Query: {query_str}

Instructions:
Extract or synthesize the answer strictly from the provided context.
Keep the answer concise and factual.
Avoid phrases like \The context states that..." | just give the answer.

assistant:
% Final answer derived solely from context.
```

**Figure 16:** Final Answer Synthesis Prompt Template. This template enforces faithful response generation based exclusively on retrieved context, a core principle in Retrieval-Augmented Generation (RAG) systems. It suppresses model hallucination by explicitly forbidding the use of prior knowledge.

```
role="system"
You are an expert tasked with evaluating two answers to the same question
based on three criteria: Comprehensiveness, Diversity, and Empowerment.

Evaluation Criteria:
· Comprehensiveness:
How much detail does the answer provide to cover all aspects
and sub-questions implied by the original query?
· Diversity:
How varied and rich is the answer in providing different perspectives,
evidence sources, or reasoning paths?
· Empowerment:
How well does the answer help the reader understand the topic
and make informed judgments or decisions?

Instructions:
Compare Answer 1 and Answer 2 for each criterion.
Choose the better answer and explain why.
Select an overall winner based on balance across all three.

Input:
Question: {query}
Answer 1: {answer1}
Answer 2: {answer2}

Output Format (JSON):
{
  "Comprehensiveness": {
    "Winner": "Answer 1 or Answer 2",
    "Explanation": "..."
  },
  "Diversity": {
    "Winner": "Answer 1 or Answer 2",
    "Explanation": "..."
  },
  "Empowerment": {
    "Winner": "Answer 1 or Answer 2",
    "Explanation": "..."
  },
  "Overall Winner": {
    "Winner": "Answer 1 or Answer 2",
    "Explanation": "..."
  }
}
```

**Figure 17:** Answer Evaluator Prompt Template. This template guides a dedicated agent to compare two candidate responses along three dimensions: comprehensiveness, diversity, and empowerment, promoting high-quality, informative, and user-centered answer selection in multi-agent systems.

