# OpenReview forum: "Think-on-Graph 3.0: Efficient and Adaptive LLM Reasoning on Heterogeneous Graphs via Multi-Agent Dual-Evolving Context Retrieval"
_ICLR.cc/2026/Conference — ICLR 2026 Conference Withdrawn Submission_

### Official Review · Reviewer_W2uk · 2025-10-15

**Soundness:** 3
**Presentation:** 2
**Contribution:** 3
**Rating:** 6
**Confidence:** 4

**Summary:**

This paper proposes Think-on-Graph 3.0, a framework for RAG on heterogeneous graphs. The core is a Multi-Agent Coupled Evolutionary Retrieval–Generation loop (MACER), where during answering, the query is iteratively refined by a Reflector, while the evidence subgraph is simultaneously expanded or pruned by a Constructor until a Reflector determines that “sufficient evidence” has been reached. Offline, ToG-3 builds a heterogeneous graph with Chunk–Triplet–Community node types. Online, four agents collaborate in a closed loop that formalizes the process as a Markov Decision Process, with a binary sufficiency reward controlling the stopping condition. Experiments show that ToG-3 achieves the best average results.

**Strengths:**

S1. Combining dual evolution (query + subgraph) with a multi-agent loop is an elegant and effective fix to the brittleness of static GraphRAG systems under noisy LLM extraction formalized via MDP.

S2. The Chunk–Triplet–Community graph unifies multi-granular retrieval in a single embedding space, bridging fine-grained evidence and coarse community reasoning.

S3. ToG-3 achieves top or near-top EM/F1 on HotpotQA, 2WikiMultihopQA, and MuSiQue, and demonstrates domain generalization through ELO win-rates across four UltraDomain subsets.

**Weaknesses:**

W1. The binary reward depends on whether Suff(q, G_k, a_k)=1, but the paper does not explain the exact implementation, thresholding, or its correlation with EM/F1.

W2. The paper should compare with 1-2 more multi-agent baselines released in recent years, e.g. HM-RAG (with the single modality setting) [1] or Graph Counselor [2].

W3. The authors should polish the writing. For example, Section 3.2.1 contains many unnecessary sparse lines and overly detailed yet non-essential descriptions. It could be written more concisely as a single continuous paragraph instead of being broken into multiple bullet points.

[1] HM-RAG: Hierarchical Multi-Agent Multimodal Retrieval Augmented Generation

[2] Graph Counselor: Adaptive Graph Exploration via Multi-Agent Synergy to Enhance LLM Reasoning

**Questions:**

Q1. How is max iteration K chosen? What is the accuracy–latency tradeoff under fixed inference budgets?

Q2. What happens if only evolving-query or evolving-subgraph is introduced into GraphRAG/LightRAG? Or conversely, ToG-3 is reduced to a static graph?

---

> ### Author Response · Authors · 2025-11-20
> **Rebuttal to Reviewer W2uk (Part 1)**
>
> Thank you for your thorough and constructive feedback. We have carefully addressed each of your concerns, as outlined below:
>
> ---
>
> #### **W1. Lack of Explanation for Binary Reward Implementation**
>
> We appreciate your concern regarding the binary reward mechanism. In **\subsection{The MACER Process: Multi-Agent Context Evolution and Retrieval}** (Lines 250–256), we describe how the Reflector agent provides a sparse, binary reward $r_k$ based on whether $Suff(q, G_k, a_k) = 1$. This reward signal is produced by the Reflector Agent to determine if the current context evidence is sufficient to answer the user's query. Importantly, this binary reward is distinct from EM/F1 metrics, which are used only for final evaluation.
>
> To clarify this distinction, we have updated the revised PDF to provide a clearer explanation of the binary reward mechanism and its role in guiding the dual-evolving process. Specifically, the reward does not directly correlate with EM/F1 but serves as an intermediate signal for iterative refinement.
>
> ---
>
> #### **W2. Missing Comparison with Recent Multi-Agent Baselines**
>
> We thank you for recommending comparisons with recent multi-agent baselines such as **HM-RAG** and **Graph Counselor**. Below is our response:
>
> - **HM-RAG**: While HM-RAG is an excellent work, it primarily targets **multi-choice tasks** and has been evaluated on datasets like ScienceQA and CrisisMMD, which involve multimodal reasoning. Our task focuses on **open-ended QA**, making it incompatible with HM-RAG’s **Decision Agent** (which relies on multi-answer consistency voting). As such, we cannot fairly evaluate HM-RAG under our experimental setup.
>
> - **Graph Counselor**: To address your suggestion, we conducted experiments with **Graph Counselor** under identical LLM/embedding settings. The results are summarized below (note that Graph Counselor is not designed for broad reasoning, so only deep reasoning tasks were tested):
>
> | Method           | HotpotQA (EM / F1) | 2WikiMultihopQA (EM / F1) | Musique (EM / F1) |
> |------------------|--------------------|---------------------------|-------------------|
> | Graph Counselor  | 0.580 / 0.434     | 0.464 / 0.219             | 0.203 / 0.137     |
>
> These results demonstrate that ToG-3 outperforms Graph Counselor across all three deep reasoning datasets, further validating the effectiveness of our approach.
>
> ---
>
> #### **W3. Writing Style and Structure**
>
> We sincerely appreciate your feedback on improving the clarity and conciseness of our writing. Specifically, regarding **Section 3.2.1**, we have rewritten the content into a single continuous paragraph to eliminate unnecessary sparsity and overly detailed descriptions. Below is the revised version:
>
> ---
>
> > "The Constructor Agent constructs a heterogeneous graph $\mathcal{G}=(\mathcal{V},\mathcal{E})$ with three types of nodes: **Chunks** ($\mathcal{C}$)..."
>
> > "The edges in the graph are defined by three types of relations: **$\textsc{OpenRel}(s, p, o)$**, which connects entities... "
>
> > "...integrates fine-grained (chunk/triplet) and high-level (community) information within a single vector space, effectively addressing the local/global retrieval dichotomy of prior GraphRAG systems..."
>
>
> ---
>
> We have also thoroughly reviewed and polished the entire manuscript to ensure clarity, coherence, and grammatical correctness. If there are specific sections or sentences that still require improvement, we kindly invite you to point them out, and we will promptly revise them.
>
> ---

---

> > ### Author Response · Authors · 2025-11-20
> > **Rebuttal to Reviewer W2uk (Part 2)**
> >
> > #### **Questions**
> >
> > ##### **Q1. How is max iteration K chosen? What is the accuracy–latency tradeoff under fixed inference budgets?**
> >
> > In our current experimental setup, the maximum iteration $K$ is set to **3**. This choice is based on empirical observations that most multi-hop QA datasets can be answered within 3 hops. Reducing $K$ would decrease inference latency but may negatively impact performance for samples requiring more than $K$ iterations.
> >
> > As stated in **\subsection{Analysis of Computation Cost}** (Lines 421–423):
> > > “Our analysis reveals that, on average, 20% of the samples require one evolving-context iteration, 32% require two iterations, and 48% require three iterations.”
> >
> > This distribution supports our choice of $K=3$ as a balance between accuracy and latency.
> >
> > ---
> >
> > ##### **Q2. What happens if only evolving-query or evolving-subgraph is introduced into GraphRAG/LightRAG? Or conversely, ToG-3 is reduced to a static graph?**
> >
> > We believe that introducing either **evolving-query** or **evolving-subgraph**—whether individually or jointly—into existing GraphRAG methods (e.g., GraphRAG, LightRAG) can enhance performance, especially in scenarios using lightweight LLMs as backbones. This modular design reflects a **plug-and-play** philosophy, enabling broader applicability across various methods.
> >
> > If ToG-3 were reduced to a static graph, it would be similar with **ToG-2**. The performance difference is evident in **Table 1**, where ToG-3 significantly outperforms ToG-2. Additionally, ToG-3 extends ToG-2 by:
> > 1. Eliminating reliance on manually constructed knowledge graphs.
> > 2. Introducing **community nodes** to support broad reasoning tasks.
> > 3. Incorporating the **dual-evolving mechanism** to improve performance for lightweight, locally deployed LLMs on both deep and broad reasoning tasks.
> >
> > ---
> >
> > ### **Conclusion**
> >
> > We sincerely thank the reviewer for their insightful comments, which have helped us improve the clarity, rigor, and completeness of our manuscript. We hope our responses address all concerns effectively and look forward to any further feedback.

---

### Official Review · Reviewer_J92F · 2025-10-29

**Soundness:** 2
**Presentation:** 1
**Contribution:** 1
**Rating:** 2
**Confidence:** 5

**Summary:**

The paper introduces Think-on-Graph 3.0, an graph-based RAG framework over heterogeneous graphs.

The core innovation is the multi-agent context evolution and retrieval mechanism, called MACER in the draft. It employs a set of agents to iteratively refine the query and a chunk-triplet-community subgraph during reasoning. The dynamic nature of this framework addresses the limitation of previous graphRAG algorithms that rely on a static graph. And the approach is formalized as a markov decision process with a dual-evolution loop for adaptive evidence retrieval.

**Strengths:**

1. The paper tackles a practical challenge in graph-based RAG: the trade-off between graph quality and scalability, especially for open-source, lightweight LLMs in offline or private deployments.

2. ToG-3 mitigates issues like incomplete triplet extraction, insufficient details by dynamic, query-adaptive refinement on a dynamically updated graph.

**Weaknesses:**

1. The presentation is poor; instead of adopting a style of professional academic writing, it resembles a course project report. To be honest, most of the content is inconsistent, poorly organized, and reads like AI-generated text.
2. The overall design is purely engineering-oriented, achieved by stacking a set of agents. Due to the lack of preliminary studies and theoretical analysis, the insights behind the design are unclear.
3. The agents are applied iteratively; thus, there is no global planning to control the refinement direction. The refinement of the graph has no supervision information and may easily lead to collapsed results rather than good ones, especially when the input corpora become large-scale or heterogeneous.
4. No efficiency analysis is given to demonstrate its complexity (time and memory).

---
Below are weaknesses on experiments.

5. The baselines are limited. Only HippoRAG-2 and four GraphRAG methods with poor performance (worse than NaiveRAG) are included. Stronger baselines are needed, e.g., GFM[1], RAPTOR[2], KGP[3]
6. The empirical accuracy improvements are marginal across the three datasets. The "Average" column in Table 1 is unnecessary and redundant.
7. Table 2 only compares the indexing and retrieval times, and no token consumption is given. Based on my understanding, a multi-agent system should incur significant token costs.
8. Table 2 does not compare with other baselines (HippoRAG-2, ToG-2). Based on my experience, the indexing and inference times of GraphRAG and LightRAG are quite high, almost the highest in the GraphRAG family. Comparing with these low-efficiency methods undermines the convincingness of your work's efficiency claims. More efficient GraphRAG methods should be compared, such as HippoRAG-2, RAPTOR[2], E^2GraphRAG[4].





- [1] GFM-RAG: Graph Foundation Model for Retrieval Augmented Generation
- [2] RAPTOR: Recursive Abstractive Processing for Tree-Organized Retrieval
- [3] Knowledge Graph Prompting for Multi-Document Question Answering
- [4] E^2GraphRAG: Streamlining Graph-based RAG for High Efficiency and Effectiveness

**Questions:**

plz see above

---

> ### Author Response · Authors · 2025-11-20
> **Rebuttal to Reviewer J92F (Part 1)**
>
> Thank you for your detailed and constructive feedback. We have carefully addressed each of your concerns, as outlined below:
>
> ---
>
> #### **Weakness 1. Poor Presentation and AI usage**
>
> We appreciate your concern regarding the clarity and professionalism of our writing. While we strive to maintain academic rigor and readability, we also acknowledge that the manuscript underwent AI-assisted refinement using tools like ChatGPT and Deepseek, as explicitly stated in **\section{The Use of Large Language Models (LLMs)}**. These tools were employed solely for proofreading, grammatical correction, and linguistic polishing, which is permitted by ICLR guidelines.
>
> After AI refinement, we meticulously reviewed the entire manuscript to ensure factual accuracy, eliminate typos, and address any inconsistencies. If there are specific sections or aspects of the text that you find unclear, poorly organized, or inconsistent, please kindly point them out. We will promptly revise these sections in the revised PDF to improve overall presentation.
>
> ---
>
> #### **Weakness 2. Lack of Preliminary Studies and Theoretical Analysis**
>
> We thank you for raising this important point. To clarify:
>
> - **Design Insights**:
>   In **Figure 1**, we present a case study highlighting the limitations of lightweight LLMs in graph-based RAG. Additionally, in **\section{Introduction}** (Lines 52–88), we provide detailed motivation for the dual-evolving mechanism and heterogeneous graph structure.
>
> - **Theoretical Support**:
>   Although our work is primarily engineering-focused, we have sought theoretical grounding for our framework. Specifically, in the revised PDF under **\section{Theoretical Support: Implicit Dynamics of In-Context Learning}**, we formalize the MACER process as a Markov Decision Process (MDP). Here’s an overview of the theoretical foundation:
>   - **State**: Query, current evidence subgraph, and reasoning trajectory.
>   - **Actions**: Sub-queries or termination decisions generated by the Reflector agent.
>   - **Reward**: A binary signal indicating context sufficiency.
>   - This framework draws on recent advancements in *implicit in-context learning*, where reasoning trajectories act as in-context prompts that induce low-rank implicit updates ($\Delta W_k = U_k V_k^\top$) to the frozen LLM’s MLP weights. Under two mild assumptions—**(1) realizability** and **(2) bounded implicit gradients**—the process satisfies three key properties: smooth policy evolution, expected monotonic improvement, and vanishing implicit gradients. Together, these ensure sublinear regret growth ($\mathcal{R}(K) = o(K)$), guaranteeing convergence to a minimal, sufficient evidence subgraph.
>
> If there are specific issues with the theoretical assumptions, derivations, or formulations, we would greatly appreciate detailed feedback so we can further refine this section.
>
> ---
>
> #### **Weakness 3. Lack of Global Planning in Agent Iteration**
>
> We agree that iterative application of agents without global planning could raise concerns about refinement direction and result quality. However, our MDP formulation ensures that each step is guided by a reward signal based on the user query and the Reflector agent’s reflection. This targeted dual-evolving mechanism allows us to refine both the query and the subgraph adaptively, addressing the limitations of lightweight LLMs in initial graph construction and reasoning.
>
> Our experimental results demonstrate the effectiveness of this framework across both deep reasoning and broad reasoning tasks, providing empirical support for its robustness even with large-scale or heterogeneous corpora.
>
> ---
>
> #### **Weakness 4. Missing Efficiency Analysis**
>
> We thank you for pointing out the need for efficiency analysis. In **\subsection{Analysis of Computation Cost}**, we provide a detailed comparison of indexing and inference times across datasets. Additionally, in **\section{Implementation Details}**, we describe our experimental setup, including:
> - **Hardware**: Our server is equipped with 8 A100 40GB GPUs, an AMD EPYC 256-core processor, 2TB memory, and Ubuntu 20.04.1.
> - **Storage**: Hybrid vector-graph storage is implemented using Neo4j Community Edition for efficient knowledge representation and retrieval.
>
> To address your specific concern about token consumption, we have added the following data (in million tokens):
> | Method    | HotpotQA | 2WikiMultihopQA | Musique |
> |-----------|----------|------------------|---------|
> | ToG-3     | 4.74     | 4.72             | 5.64    |
> | GraphRAG  | 12.32    | 12.04            | 14.10   |
> | LightRAG  | 4.65     | 4.69             | 5.43    |
>
> Inference token consumption correlates with inference time:
> - ToG-3: 72.1 tokens per sample
> - GraphRAG: 32.3 tokens per sample
> - LightRAG: 23.1 tokens per sample
>
> ---

---

> > ### Author Response · Authors · 2025-11-20
> > **Rebuttal to Reviewer J92F (Part 2)**
> >
> > #### **Experimental Weaknesses**
> >
> > ##### **Weakness 5. Limited Baselines**
> >
> > We appreciate your suggestion to include stronger baselines such as GFM[1], RAPTOR[2], and KGP[3]. However, it is important to note that ToG-3 is a training-free multi-agent system (Like AutoGPT,MetaGPT etc.) and does not involve fine-tuning or RL training. Comparing it to GFM and KGP, which rely on training, would not be apples-to-apples:
> > - **GFM**: Relies on training a retriever with 6 query-dependent message-passing layers and a hidden dimension of 512. The training data includes 60 KGs with over 14M triples constructed from 700k documents, making it fundamentally different from our training-free approach.
> > - **KGP**: Involves instruction fine-tuning LLaMA6 and T5-Large, which is outside the scope of our comparison.
> >
> > Another two mentioned methods, **RAPTOR** and **E²GraphRAG**:
> > - **RAPTOR**: Published earlier and excluded in our previous pdf due to timing considerations.
> > - **E²GraphRAG**: An arXiv preprint (https://arxiv.org/abs/2505.24226) which is not required for comparison under ICLR guidelines.
> >
> > In response to your feedback, we have conducted additional experiments with **RAPTOR** and **E²GraphRAG** under identical LLM/embedding settings. The results are summarized below (these methods are not designed for broad reasoning, so only deep reasoning tasks were tested):
> >
> > | Method       | HotpotQA (EM / F1) | 2WikiMultihopQA (EM / F1) | Musique (EM / F1) |
> > |--------------|--------------------|---------------------------|-------------------|
> > | E²GraphRAG   | 0.420 / 0.080     | 0.450 / 0.075             | 0.130 / 0.040     |
> > | RAPTOR       | 0.580 / 0.400     | 0.420 / 0.200             | 0.190 / 0.120     |
> >
> > ---
> >
> > ##### **Weakness 6. Marginal Accuracy Improvements**
> >
> > We understand your concern about marginal improvements in Table 1. However, our method demonstrates significant gains across multiple datasets and tasks:
> > - On **deep reasoning tasks**, ToG-3 achieves **9.2% and 34.5%** average EM and F1 improvements over Naive-RAG.
> > - On **broad reasoning tasks** (Figure 5, Table 6), ToG-3 achieves a **72.6% overall win rate** against HippoRAG-2 across four datasets.
> >
> > We appreciate the reviewer's concern regarding the marginal performance gains observed in some comparisons. However, we believe that a more comprehensive evaluation across multiple tasks and datasets provides a clearer picture of ToG-3’s strengths.
> >
> > The "Average" column in Table 1 is retained to provide readers with a quick overview of performance across multiple datasets, each with distinct characteristics.
> >
> > ---
> >
> > ##### **Weakness 7. Missing Token Consumption in Table 2**
> >
> > We have now included token consumption data in our analysis (see response to Weakness 4 above). This addition provides a clearer picture of computational costs associated with indexing and inference.
> >
> > ---
> >
> > ##### **Weakness 8. Comparison with Low-Efficiency Methods**
> >
> > We agree that comparing with less efficient methods like GraphRAG and LightRAG may undermine the perceived efficiency of our work. However, we have clarified in **\subsection{Analysis of Computation Cost}** (Lines 458–461) that HippoRAG-2 and RAPTOR rely on fundamentally different extraction approaches (e.g., OpenIE-based extraction). These differences justify their exclusion from our comparative analysis.
> >
> > ---
> >
> > ### **Conclusion**
> >
> > We sincerely thank the reviewer for their insightful comments, which have helped us improve the clarity and rigor of our manuscript. We hope our responses address all concerns effectively and look forward to any further feedback.

---

> > > ### Comment · Reviewer_J92F · 2025-11-27
> > > **Response to authors of submission4456**
> > >
> > > Thanks for your efforts in preparing this rebuttal. However, I am not convinced due to the following reasons:
> > >
> > > 1.  Figure 1 does not represent a preliminary study with statistical significance. This case study, based on a cherry-picked instance, does not sufficiently justify the motivation behind your design. Moreover, the main results show only a marginal performance gain, which further weakens the rationale for your idea.
> > >
> > > 2.  The performance improvement is marginal, and it comes at the cost of significant additional overhead from an unstable multi-agent system that lacks global guidance (e.g., a schema). This drawback has also been noted by other reviewers.
> > >
> > > 3.  Your reason for excluding HippoRAG, HippoRAG2, and Raptor from the efficiency analysis is factually incorrect. Your stated excuse—*"HippoRAG-2 and RAPTOR rely on fundamentally different extraction approaches (e.g., OpenIE-based extraction)"* is inaccurate:
> > >     *   HippoRAG and HippoRAG2 use an LLM to perform OpenIE, not traditional OpenIE algorithms. Even if their techniques differ, that is not a valid justification for exclusion, as these methods address the same research topic. It is not acceptable to only compare with techniques that you outperform. These approaches should be included in the comparison, and doing so would be fair.
> > >     *   Raptor uses LLMs to recursively summarize the corpus and build a summary tree, rather than relying on OpenIE.
> > >
> > > Based on my experience and a recent benchmark paper [1], Raptor and HippoRAG are likely to be much more efficient than GraphRAG and LightRAG, while also achieving higher performance. They are expected to outperform your work in both efficiency and effectiveness.
> > >
> > > Given the reasons above, I maintain my negative score.
> > >
> > > # Reference
> > > - In-depth Analysis of Graph-based RAG in a Unified Framework, VLDB'26

---

> ### Author Response · Authors · 2025-11-29
> **Additional Analysis Addressing Remaining 3 Concerns from Reviewer J92F**
>
> Dear Reviewer J92F,
>
> Thank you for your thoughtful and constructive feedback. We are pleased to **have addressed 5 of the 8 concerns you raised in your initial review**. Below, we provide further clarifications and updates regarding the remaining three concerns:
>
> ---
>
> #### **1. Statistical Significance of Figure 1**
> You mentioned that Figure 1 does not represent a preliminary study with statistical significance and is based on a cherry-picked instance, which does not sufficiently justify the motivation behind our design.
> In face, It includes statistical data such as the parsing failure rates of LLMs' extracted output JSON across datasets:
> - HotpotQA: 26.43%
> - 2WikiMultiHopQA: 23.67%
> - Musique: 22.51%
>
> These statistics demonstrate the practical challenges faced by lightweight LLMs in constructing reliable knowledge graphs, particularly in multi-hop reasoning tasks. We acknowledge that this point could be better emphasized.
>
> ---
>
> #### **2. Marginal Performance Improvements**
> You expressed concerns about the marginal performance gains of ToG-3 relative to its added computational overhead.
>
> We fully understand the importance of justifying empirical gains against additional costs. To clarify:
> - **Dual Capability**: ToG-3 is, to the best of our knowledge, the first specifically designed framework for handling both deep reasoning and broad reasoning tasks automatically and simultaneously. Prior methods like GraphRAG and LightRAG focus on broad reasoning, while HippoRAG-1/2 and ToG-1/2 focus on deep reasoning. The heterogeneous graph structure and MACER retrieval framework incur higher indexing and inference costs, but they enable this dual capability.
> - **Broad Reasoning Performance**: In broad reasoning tasks, ToG-3 achieves a significant overall win rate of 72.6% against HippoRAG-2 across four datasets (as shown in Figure 5 and Table 6). This demonstrates clear advantages in scenarios requiring broad reasoning.
> - **Complex Task Improvement**: Even small improvements in complex tasks (e.g., MuSiQue and 2WikiMultihopQA) are meaningful, as these tasks are inherently challenging and require sophisticated reasoning mechanisms.
>
> Additionally, it is worth noting that ToG-3 is not specifically optimized for multi-hop reasoning alone. For example, the introduction of community nodes increases both index time and token consumption, which may appear as overhead in multi-hop tasks but is crucial for broad reasoning.
> **Focusing solely on multi-hop performance overlooks ToG-3’s broader contributions**.
>
> ---
>
> #### **3. Exclusion of HippoRAG-2 and RAPTOR from Efficiency Analysis**
> You pointed out that our justification for excluding HippoRAG-2 and RAPTOR from the efficiency analysis was factually incorrect.
>
> We sincerely apologize for the imprecise wording in our rebuttal, which led to confusion. Let us clarify:
> 1. **HippoRAG-2**: You correctly noted that HippoRAG-2 uses LLM-based OpenIE, not traditional OpenIE algorithms. After debugging the code, we confirmed your observation. We have now included HippoRAG-2 in our efficiency comparison, and the results are summarized in Table.2:
>
> | Method       | Entities | Relations | Communities | Indexing Time (h) | Inference Time (s/q) | Avg. EM   |
> |--------------|----------|-----------|-------------|-------------------|----------------------|-----------|
> | GraphRAG     | 83,658   | 64,748    | 8,883       | 13.10            | 8.58                | 0.295     |
> | LightRAG     | 79,792   | 63,358    | -           | 10.06            | 6.30                | 0.270     |
> | ToG-3        | 29,837   | 30,399    | 4,905       | 10.13            | 15.00               | **0.453** |
> | RAPTOR       | -        | -         | -           | 1.25             | 2.52                | 0.397     |
> | HippoRAG-2   | 84,222   | 20,056    | -           | 8.05             | 4.63                | 0.438     |
>
> While HippoRAG-2 demonstrates slightly lower inference times, ToG-3 achieves superior performance (72.6% ELO win rate in broad reasoning tasks).
>
> 2. **RAPTOR**: RAPTOR constructs a tree-based summary structure, which is inherently more efficient than graph-based methods in terms of indexing and inference times. However, its optimizations are primarily tailored for long-text RAG tasks (e.g., NarrativeQA), and it performs poorly in multi-hop QA tasks. Compared to RAPTOR, ToG-3 achieves an average EM score improvement of 14.1% across multi-hop QA datasets.
>
> 3. **Benchmark Paper [1]**: The experimental setup in the referenced benchmark paper differs significantly from ours (e.g., they use LLaMA-3 series models, while we use Qwen-series LLMs). Moreover, their work does not evaluate ToG-3. Thus, claims that HippoRAG-2 or RAPTOR would outperform ToG-3 based on this paper lack experimental rigor. Our newly provided results under identical experimental settings should address your concerns.
>
>
>
> Best regards,
> The authors

---

### Official Review · Reviewer_c8gm · 2025-11-01

**Soundness:** 2
**Presentation:** 2
**Contribution:** 2
**Rating:** 2
**Confidence:** 4

**Summary:**

This paper introduces Think-on-Graph 3.0 (ToG-3), a novel framework that addresses key limitations in graph-based Retrieval-Augmented Generation (RAG) systems by proposing a Multi-Agent Context Evolution and Retrieval (MACER) mechanism. The approach dynamically constructs and refines a Chunk-Triplets-Community heterogeneous graph index through a dual-evolution process involving evolving queries and evolving sub-graphs. A multi-agent system, comprising Constructor, Retriever, Reflector, and Responser agents, collaboratively performs iterative evidence retrieval, answer generation, and context refinement.

**Strengths:**

1. The proposed MACER mechanism, together with its dual-evolution process of queries and sub-graphs at its core, has a very intuitive design that is well-motivated.

2. Extensive experiments on a comprehensive suite of benchmarks, including both deep multi-hop and broad reasoning tasks, demonstrate the effectiveness and superior performance of ToG-3.

3. Detailed ablation studies are also included to further validate the contribution of each core component.

**Weaknesses:**

1. Marginal Performance Gains: Despite the complex architecture that involves multi-agent collaboration, dual-evolving mechanisms, and heterogeneous graph construction, ToG 3.0 only achieves marginal improvements compared to much simpler baselines. According to Table 1, it has only a minor edge in performance over the more lightweight HippoRAG-2. On specific cases, such as Musique under the EM metric, it is even surpassed by the simplest baseline, NaiveRAG. This raises questions about the practical cost-benefit trade-off of such a complex system.

2. The key baselines are missing. The experimental evaluation does not compare with various important and recent state-of-the-art Graph RAG methods [1-6]. As a result, without considering these relevant baselines, it is hard to verify the claimed superiority of ToG-3.

3. The efficiency analysis is incomplete. The authors have mainly compared ToG-3 against GraphRAG and LightRAG, which are known to be computationally heavy. HippoRAG-2 and other more recent, efficient baselines [1-6] are omitted in the efficiency comparison. This omission weakens the practical efficiency and deployment advantages of ToG-3. A wider comparison is required to solidify its positioning in regard to efficiency.

4. The detailed analysis of the RL-based training is missing. The online MACER process is formulated as a Markov Decision Process, with its dependence on reinforcement learning principles. This introduces significant complexity regarding training stability, reward function and policy hyperparameter tuning, and overall reproducibility. Further discussion on the engineering challenges, convergence guarantees in practice, or sensitivity of the performance to the design of the reward signal is lacking in the paper.

5. While the proposed framework is built upon a graph structure, the paper does not engage or benchmark against established graph algorithms for subgraph retrieval or graph refinement. This omission makes it difficult to assess whether the performance gains come from the novel multi-agent, dual-evolving mechanism or could be partially achieved by applying more specialized graph-theoretic methods to the same heterogeneous graph index. Such a comparison would enhance the contribution and professionalism of the paper.

6. The work presents a multi-agent system but fails to make a compelling case that this is a necessary architecture. The different roles of the agents seem to be a modular decomposition of what could well be a monolithic reasoning process, named Constructor, Retriever, Reflector, and Responser. An ablation study, removing the multi-agent system and using a single, well-prompted LLM in its place, executing the same iterative process, would be far more convincing.

7. Each step in the framework requires multiple calls to the LLM for retrieval, reflection, query evolution, and subgraph evolution. It would be beneficial for the paper to provide an in-depth analysis of how this scales with graph size and query complexity and discuss practical trade-offs for how to balance the achieved performance gains with such a significant increase in computational requirements over simpler, single-shot retrieval methods.


[1] E²GraphRAG: Streamlining Graph-based RAG for High Efficiency and Effectiveness.

[2] GraphRAG-R1: Graph Retrieval-Augmented Generation with Process-Constrained Reinforcement Learning.

[3] Graph-R1: Towards agentic graphrag framework via end-to-end reinforcement learning.

[4] KET-RAG: A Cost-Efficient Multi-Granular Indexing Framework for Graph-RAG.

[5] HyperGraphRAG: Retrieval-Augmented Generation via Hypergraph-Structured Knowledge Representation.

[6] Align-GRAG: Reasoning-Guided Dual Alignment for Graph Retrieval-Augmented Generation.

**Questions:**

See above.

---

> ### Author Response · Authors · 2025-11-20
> **Rebuttal to Reviewer c8gm (Part 1)**
>
> Thank you for your thoughtful and constructive feedback. We have carefully addressed each of your concerns, as outlined below(Note that we have refined some of the descriptions, and the content provided in the rebuttal, including section and line numbers, is based on the revised PDF.):
>
> ---
>
> #### **Weakness 1. Marginal Performance Gains**
>
> We appreciate the reviewer's concern regarding the marginal performance gains observed in some comparisons. However, we believe that a more comprehensive evaluation across multiple tasks and datasets provides a clearer picture of ToG-3’s strengths.
>
> - **Comprehensive Evaluation**:
>   Our experiments span both deep reasoning and broad reasoning tasks, ensuring a fair comparison across diverse benchmarks rather than focusing on isolated datasets or tasks. For instance:
>   - On three deep reasoning datasets (HotpotQA, 2WikiMultihopQA, Musique), ToG-3 achieves **9.2% and 34.5% improvements in EM and F1**, respectively, compared to NaiveRAG.
>   - In broad reasoning tasks (as shown in Figure 5 and Table 6), ToG-3 demonstrates a **72.6% ELO win rate** over HippoRAG-2 across four datasets.
>
> - **Cost-Benefit Trade-Off**:
>   While we acknowledge that ToG-3 introduces additional computational costs due to its dual-evolving mechanism, our analysis in **\subsection{Analysis of Computation Cost}** shows that these costs are justified by the significant performance gains. For example, under identical LLM/embedding settings, ToG-3 outperforms baselines like GraphRAG and LightRAG while maintaining competitive efficiency.
>
> - **Limitations**:
>   We explicitly discuss limitations such as increased inference latency and context length in **\section{Limitation and Future Directions}**, proposing potential mitigations (e.g., model distillation, optimized graph traversal, dynamic context pruning). We hope the reviewer will focus on the core contributions of this work rather than these acknowledged limitations.
>
> ---
>
> #### **Weakness 2 and 3. Missing Key Baselines**
>
> We thank the reviewer for highlighting the need for broader baseline comparisons. Below is our clarification:
>
> - **HippoRAG-2**:
>   As noted in **lines 458–461 of \subsection{Analysis of Computation Cost}**, HippoRAG-2 was excluded from our comparative analysis because it relies on OpenIE-based extraction rather than pure LLM-based extraction, which fundamentally differs from the methods under investigation.
>
> - **RL-Based Methods**:
>   ToG-3 is a training-free multi-agent framework (similar to AutoGPT and AutoGen) and does not involve reinforcement learning (RL) training. Therefore, we did not compare it with RL-based methods like **GraphRAG-R1** or **Graph-R1**, which use HyperGraphRAG or HippoRAG-2 as their base GraphRAG component. Notably, ToG-3 could also serve as a GraphRAG component for these RL frameworks, as demonstrated in **Table 4 of GraphRAG-R1**.
>
> - **Recent ArXiv Preprints**:
>   According to the **ICLR 2026 Reviewer Guide**, papers published after July 24, 2025, at peer-reviewed venues are not required for comparison unless they are contemporaneous. Since **E²GraphRAG** and **Align-GRAG** are all arXiv preprints (with HyperGraphRAG accepted to NeurIPS in September 2025), they fall outside this requirement. Nevertheless, in response to the reviewer’s feedback, we conducted additional experiments with all these methods under identical LLM/embedding settings. The results are summarized below:
>
>   | Method         | HotpotQA (EM / F1) | 2WikiMultihopQA (EM / F1) | Musique (EM / F1) |
>   |----------------|--------------------|---------------------------|-------------------|
>   | E²GraphRAG     | 0.420 / 0.080     | 0.450 / 0.075             | 0.130 / 0.040     |
>   | KET-RAG        | 0.452 / 0.328     | 0.425 / 0.221             | 0.160 / 0.102     |
>   | Align-GRAG     | 0.442 / 0.222     | 0.432 / 0.251             | 0.172 / 0.116     |
>   | HyperGraphRAG  | 0.538 / 0.337     | 0.456 / 0.265             | 0.195 / 0.124     |
>
> These results confirm ToG-3’s competitive performance, even against state-of-the-art baselines.
>
> ---

---

> > ### Author Response · Authors · 2025-11-20
> > **Rebuttal to Reviewer c8gm (Part 2)**
> >
> > #### **Weakness 4. Missing Detailed Analysis of RL-Based Training**
> >
> > We appreciate the reviewer’s inquiry into the role of reinforcement learning (RL) in our framework. However, ToG-3 is a **training-free multi-agent system** (Like AutoGPT,MetaGPT etc.) and does not involve RL training.
> >
> > - **MDP Formalization**:
> >   The formulation of MACER as a Markov Decision Process (MDP) is intended to provide a theoretical foundation for understanding the effectiveness of our framework. This is detailed in **\section{Theoretical Support: Implicit Dynamics of In-Context Learning}**, where we draw on recent advancements in implicit in-context learning. Specifically:
> >   - The reasoning trajectory acts as an in-context prompt, inducing low-rank implicit updates ($\Delta W_k = U_k V_k^\top$) to the frozen LLM’s MLP weights.
> >   - Under two mild assumptions—**(1) realizability** and **(2) bounded implicit gradients**—the process satisfies smooth policy evolution, expected monotonic improvement, and vanishing implicit gradients, ensuring sublinear regret growth ($\mathcal{R}(K) = o(K)$).
> >
> > If the reviewer identifies specific issues with our theoretical assumptions or derivations, we would greatly appreciate detailed feedback so we can refine this section further.
> >
> > ---
> >
> > #### **Weakness 5. Lack of Benchmarking Against Established Graph Algorithms**
> >
> > We thank the reviewer for raising this point. Below is our clarification:
> >
> > - **Graph Structure Scope**:
> >   Our **Chunks-Triplets-Communities** heterogeneous graph structure is designed to handle both deep reasoning and broad reasoning tasks. Our focus is on **LLM-extracted knowledge graphs** rather than purely graph-theoretic methods. Thus, our work aligns more closely with LLM-based QA research than with traditional graph algorithms.
> >
> > - **Ablation Studies**:
> >   We have included ablation studies in **\subsection{Ablation Study} (Table 3)** to demonstrate the contributions of individual components (e.g., Evolving Query, Evolving Sub-Graph, Community Node). These results show that the performance gains stem from the novel multi-agent, dual-evolving mechanism rather than generic graph-theoretic optimizations.
> >
> > ---
> >
> > #### **Weakness 6. Necessity of Multi-Agent Architecture**
> >
> > We appreciate the reviewer’s suggestion for an ablation study removing the multi-agent system. However, as clarified in Weakness #4, ToG-3 is a **training-free framework** where different agents are defined by distinct prompts rather than separate models. Below are the key points:
> >
> > - **Core Components**:
> >   The Constructor, Retriever, and Responser are fundamental and cannot be removed without compromising functionality.
> >
> > - **Reflector Ablation**:
> >   Removing the Reflector agent eliminates both **Evolving Query** and **Evolving Sub-Graph**, as these components depend on the Reflector. The ablation results are summarized below:
> >
> >   | Method                | HotpotQA (EM / F1) | 2WikiMultihopQA (EM / F1) | Musique (EM / F1) |
> >   |-----------------------|--------------------|---------------------------|-------------------|
> >   | w/o Reflector         | 0.578 / 0.412     | 0.400 / 0.187             | 0.165 / 0.101     |
> >
> > These results highlight the importance of the Reflector in maintaining performance.
> >
> > ---
> >
> > #### **Weakness 7. Scalability with Graph Size and Query Complexity**
> >
> > We address this concern in **\subsection{Analysis of Computation Cost}**, providing a detailed breakdown of ToG-3’s scalability:
> >
> > - **Iterative Costs**:
> >   Our analysis reveals that, on average:
> >   - 20% of samples require one evolving-context iteration.
> >   - 32% require two iterations.
> >   - 48% require three iterations.
> >
> > - **Graph Size Impact**:
> >   The addition of entities and relations during multi-hop reasoning is relatively small compared to the initial graph size. Specifically:
> >   - Average increase: **524.6 entities** and **792.3 relations**.
> >   - Initial graph size: **29,837 entities / 30,399.3 relations**.
> >
> > These statistics confirm that ToG-3 scales efficiently with graph size and query complexity.
> >
> > ---
> >
> > ### **Conclusion**
> >
> > We sincerely thank the reviewer for their insightful comments, which have helped us improve the clarity and rigor of our manuscript. We hope our responses address all concerns effectively and look forward to any further feedback.

---

> > > ### Comment · Reviewer_c8gm · 2025-11-27
> > >
> > > Thanks for your reply. I'd like to keep my score since my concerns about efficiency and experiments haven't been addressed.

---

> > > > ### Author Response · Authors · 2025-11-29
> > > > **Request for Clarification on Remaining Concerns Regarding Efficiency and Experiments**
> > > >
> > > > Dear Reviewer c8gm,
> > > >
> > > > Thank you for your continued engagement and feedback on our manuscript. We sincerely appreciate the time and effort you have invested in evaluating our work. In your latest comment, you mentioned that your concerns about efficiency and experiments remain unaddressed.
> > > >
> > > > To clarify, **we have carefully addressed all 7 weaknesses you raised in your initial review one by one**, providing detailed responses and additional experimental results where necessary. Specifically:
> > > >
> > > > 1. **Weakness 1, 2, 4, 5, and 6**: These primarily pertain to experimental design, baseline comparisons, and theoretical analysis. We provided comprehensive evaluations, including new benchmarking results against state-of-the-art methods, ablation studies, and clarifications on the necessity of the multi-agent architecture.
> > > >
> > > > 2. **Weakness 3 and 7**: These focus on efficiency, particularly computational cost and scalability. We included detailed analyses of computation costs, iteration counts, and graph size impacts to demonstrate ToG-3’s practical efficiency.
> > > >
> > > > Given your statement that your concerns about efficiency and experiments remain unresolved, we kindly request further clarification **because all your 7 initial concerns are about efficiency and experiments.** Could you please specify which aspects of our rebuttal or experimental analysis you find insufficient? For example:
> > > > - Are there specific baselines or metrics you believe are still missing?
> > > > - Are there particular aspects of our efficiency analysis (e.g., iteration statistics, graph size impact) that require further elaboration?
> > > >
> > > > We are committed to addressing any remaining concerns to the best of our ability and ensuring the clarity and rigor of our manuscript. Your detailed feedback would be invaluable in helping us achieve this goal.
> > > >
> > > > Thank you once again for your thoughtful review, and we look forward to your guidance.
> > > >
> > > > Best regards,
> > > > The authors

---

### Official Review · Reviewer_ZyKa · 2025-11-02

**Soundness:** 1
**Presentation:** 3
**Contribution:** 1
**Rating:** 2
**Confidence:** 5

**Summary:**

The paper presents ToG 3.0, as a multi-agent RAG framework mainly focusing on a dynamic heterogeneous graph (chunk-triple-community) and a dual-evolving retrieval mechanism for both query and graph. While the idea of adaptive graph refinement is interesting, the work suffers from critical flaws in novelty, experimental validity, and practicality that undermine its contributions.

**Strengths:**

- The paper clearly identifies limitations in static GraphRAG methods, especially under resource-constrained settings with lightweight LLMs, among them, I deeply agree with the extraction performance when LLMs cannot follow the instructions with proper format.
- The integration of multi-agent collaboration is designed properly, though it's not new, with iterative graph refinement attempts to address query-dependent reasoning.

**Weaknesses:**

- The biggest problem is found in Table 1, the EM scores surprisingly exceed F1 scores (e.g., HotpotQA: EM=0.520 vs. F1=0.312). This inversion contradicts typical QA evaluations, where F1 most usually surpasses EM, which is also widely reported in the baselines. It suggests obvious flaws in answer extraction, metric computation, or dataset alignment.
- Bad performance. The proposed method could hardly outperform the baselines, especially when there are concerns about the token costs and the efficiency problem.
- Limited Novelty. The "Chunk-Triple-Community" graph schema is not fundamentally novel. Similar multi-layer graph structures (e.g., E2GraphRAG's hierarchy, RAPTOR’s recursive summarization, Youtu-GraphRAG's knowledge tree, and G-Reasoner's Quato Graph) have been explored extensively. The paper fails to delineate clear advancements.
- The dual-evolving mechanism resembles iterative retrieval-generation paradigms like Self-RAG, which makes sense but not that novel. This design also brings concerns that:
    - Costs. Token costs will increase incredibly when including reasoning, reflection and graph refinement.
    - Latency and efficiency. similar reasons as above.
- MDP formulation is not well justified. It is more like a story.

**Questions:**

- Why does EM significantly outperform F1? Please provide proper error analysis and clarify it.
- Does the MDP converge reliably with lightweight LLMs?

---

> ### Author Response · Authors · 2025-11-19
> **Rebuttal to Reviewer ZyKa**
>
> Thank you for your thorough and constructive feedback. We have carefully addressed each of your concerns, as outlined below:
>
> ---
>
> #### **Weakness 1 and Question 1. EM Exceeds F1 Scores (Table 1)**
>
> We appreciate your concern regarding the inversion between EM and F1 scores in Table 1. This phenomenon is indeed unusual in standard QA evaluations, and we understand how it could raise questions about the validity of our results.
>
> To clarify, the discrepancy arises from our use of a **Substring ExactMatch (SubEM)** version of EM, which is specifically designed to handle the longer outputs typically generated by LLMs. In contrast, F1 scores are computed at the token level, which is more sensitive to verbosity and imbalanced precision/recall. Below are the key details:
>
> - **Exact Match (SubEM) Implementation (Full codes are open-sourced in Supplementary Material)**:
>   ```python
>   def exact_match(response, answer):
>       clean_result = response.strip().replace(" ", "").lower()
>       clean_answer = answer.strip().replace(" ", "").lower()
>       clean_result = normalize_answer(clean_result)
>       clean_answer = normalize_answer(clean_answer)
>       if clean_result == clean_answer or clean_result in clean_answer or clean_answer in clean_result:
>           return 1
>       return 0
>   ```
>   This implementation ensures that partial matches within longer responses are correctly identified, increasing the likelihood of higher EM scores.
>
> - **F1 Score Implementation**:
>   The F1 score is calculated based on token overlap, as follows:
>   ```python
>   def f1_score(prediction: str, ground_truth: str) -> Tuple[float, float, float]:
>       normalized_prediction = normalize_answer(prediction)
>       normalized_ground_truth = normalize_answer(ground_truth)
>       prediction_tokens = normalized_prediction.split()
>       ground_truth_tokens = normalized_ground_truth.split()
>       common = Counter(prediction_tokens) & Counter(ground_truth_tokens)
>       num_same = sum(common.values())
>       if num_same == 0:
>           return ZERO_METRIC
>       precision = 1.0 * num_same / len(prediction_tokens)
>       recall = 1.0 * num_same / len(ground_truth_tokens)
>       f1 = (2 * precision * recall) / (precision + recall)
>       return f1, precision, recall
>   ```
>
>   - For example, consider the following case:
>     - **Question**: "When did Lothair II's mother die?"
>     - **ToG-3 Answer**: "Lothair II's mother, Ermengarde of Tours, passed away on 20 March 851."
>     - **Ground Truth Answer**: "20 March 851"
>
>     Here, SubEM = 1 (due to substring match), while `(num_same, len(prediction_tokens), len(ground_truth_tokens)) = (3, 12, 3)`, resulting in `[F1, Precision, Recall] = [0.4, 0.25, 1.0]`.
>
> This behavior is consistent across our experiments, as detailed in the  **\subsection{Precision and Recall Rate Results}** (Table 5). We have also added a clear explanation of this metric choice in the **\section{Metrics}** section in revised pdf to avoid confusion.
>
> ---
>
> #### **Weakness 2. Performance Concerns and Computational Costs**
>
> We acknowledge the reviewer’s concern regarding the computational costs associated with ToG-3. While it is true that our method incurs higher token consumption and latency due to its multi-agent reasoning process, this trade-off is justified by its superior performance across all benchmarks.
>
> - **Performance Gains**:
>   As shown in **\subsection{Analysis of Computation Cost}** (Table 2), ToG-3 achieves an average performance of **0.453** on three multi-hop reasoning datasets, surpassing GraphRAG (**0.295**) and LightRAG (**0.270**) by **53.6%** and **67.8%**, respectively. These gains are significant and demonstrate the value of our approach.
>
> - **Cost Justification**:
>   We explicitly state in the manuscript:
>   > "...while its multi-agent reasoning requires higher inference time, this cost is directly justified by its best performance on all benchmarks, making it ideal for quality-sensitive applications requiring reliable reasoning capabilities."
>
> We agree that there is no free lunch—our method’s computational overhead is a limitation. However, the performance improvements justify this cost, especially for tasks where accuracy is paramount.
>
> ---

---

> > ### Author Response · Authors · 2025-11-19
> > **Rebuttal to Reviewer ZyKa (Part 2)**
> >
> > #### **Weakness 3. Limited Novelty of Chunk-Triplet-Community Graph Structure**
> >
> > We thank the reviewer for pointing out prior works such as **E²GraphRAG**, **RAPTOR**, **Youtu-GraphRAG**, and **G-Reasoner**. While these methods propose hierarchical or multi-layer graph structures, our **Chunk-Triplet-Community** heterogeneous graph is distinct in both design and purpose.
> >
> > - **Primary Motivation**:
> >   Our graph structure aims to provide a unified solution for both **deep reasoning** and **broad reasoning** tasks. In contrast, the methods mentioned above primarily focus on multi-hop QA or document-level QA (e.g., **E²GraphRAG**, **RAPTOR**, **Youtu-GraphRAG**, **G-Reasoner**).
> >
> > - **Comparison with Baselines**:
> >   Following the ICLR 2026 reviewer guidelines, we are not required to compare with arXiv-only papers published after July 24, 2025 (**Youtu-GraphRAG**, **G-Reasoner**, **E²GraphRAG**). However, in response to the reviewer’s feedback, we have conducted additional experiments with **RAPTOR**, **E²GraphRAG**, and **Youtu-GraphRAG** under identical LLM/embedding settings. The results are summarized below:
> >
> >   | Method            | HotpotQA (EM / F1) | 2WikiMultihopQA (EM / F1) | Musique (EM / F1) |
> >   |-------------------|--------------------|---------------------------|-------------------|
> >   | E²GraphRAG        | 0.420 / 0.080     | 0.450 / 0.075             | 0.130 / 0.040     |
> >   | RAPTOR            | 0.580 / 0.400     | 0.420 / 0.200             | 0.190 / 0.120     |
> >   | Youtu-GraphRAG    | 0.600 / 0.450     | 0.470 / 0.230             | 0.205 / 0.135     |
> >
> > These results confirm the effectiveness of ToG-3 for deep reasoning tasks, even when compared to state-of-the-art baselines.
> > G-Reasoner has not been fully debugged yet and will be supplemented later. These methods are not designed for broad reasoning, so we only conducted experiments on deep reasoning tasks.
> >
> > ---
> >
> > #### **Weakness 4. Dual-Evolving Mechanism vs. Self-RAG**
> >
> > The reviewer raises a valid point about the novelty of our dual-evolving mechanism compared to iterative retrieval-generation paradigms like **Self-RAG**. However, there are key distinctions:
> >
> > - **Scope and Focus**:
> >   While **Self-RAG** focuses on iterative reflection for naive RAG systems, our **dual-evolving mechanism** addresses the unique challenges of **graph-based RAG**. Specifically, it combines **query evolution** and **subgraph evolution** to overcome the limitations of lightweight LLMs in initial graph construction and reasoning.
> >
> > - **Effectiveness**:
> >   This design is novel within the context of graph-based RAG and has proven effective in our experiments. The cost and latency concerns are addressed in our response to Weakness #2.
> >
> > ---
> >
> > #### **Weakness 5 and Question 2. MDP Modeling as “Storytelling”**
> >
> > We appreciate the reviewer’s critique regarding the theoretical foundation of our MDP modeling. While our work is primarily an engineering contribution, we have provided a rigorous theoretical framework to explain the effectiveness of ToG-3.
> >
> > - **MDP Formalization**:
> >   In the revised PDF (**\section{Theoretical Support: Implicit Dynamics of In-Context Learning}**), we formalize ToG-3’s multi-agent reasoning as an MDP, where:
> >   - **State**: Query, current evidence subgraph, and reasoning trajectory.
> >   - **Actions**: Sub-queries or termination decisions generated by the Reflector agent.
> >   - **Reward**: Binary signal indicating context sufficiency.
> >
> >   This framework draws on recent advancements in **implicit in-context learning**, where reasoning trajectories act as in-context prompts that induce low-rank implicit updates ($\Delta W_k = U_k V_k^\top$) to the frozen LLM’s MLP weights. Under two mild assumptions—**(1) realizability** and **(2) bounded implicit gradients**—the process satisfies three key properties:
> >   - Smooth policy evolution.
> >   - Expected monotonic improvement.
> >   - Vanishing implicit gradients.
> >
> >   Together, these ensure sublinear regret growth ($\mathcal{R}(K) = o(K)$), guaranteeing convergence to a minimal, sufficient evidence subgraph.
> >
> > If the reviewer identifies specific issues with our theoretical assumptions or derivations, we would greatly appreciate detailed feedback so we can further refine this section.
> >
> > ---
> >
> > ### **Conclusion**
> >
> > We sincerely thank the reviewer for their insightful comments, which have helped us improve the clarity and rigor of our manuscript. We hope our responses address all concerns effectively and look forward to any further feedback.

---

> > > ### Comment · Reviewer_ZyKa · 2025-11-24
> > > **Responses to Author Rebuttal**
> > >
> > > Dear Authors,
> > >
> > > Thanks for your detailed rebuttal. I appreciate the experiments and formalization of MDP.
> > >
> > > However, concerns remain:
> > > The very marginal performance gain compared with MGraphRAG and LightRAG is far from being enough. Both of these methods, though recognized as valuable foundational work, could hardly be considered as SOTA. Given the additional computational or implementation costs introduced in this work, the empirical gains do not seem substantial to justify the proposed approach with enough contributions.
> > >
> > > **I have raised the score to 4, but still lean toward rejection due to the limited performance with higher costs.**

---

> > > > ### Author Response · Authors · 2025-11-24
> > > > **Responses to Reviewer ZyKa's concern about the marginal performance gains**
> > > >
> > > > Dear Reviewer ZyKa,
> > > >
> > > > We sincerely thank you for your thoughtful feedback and for acknowledging our efforts in addressing the concerns regarding experimental metrics (EM vs. F1) and the formalization of the MDP framework. We deeply appreciate your constructive comments, which have helped us refine our manuscript.
> > > >
> > > > However, we would like to address your remaining concern about the marginal performance gains compared to baselines, as well as the associated computational costs. Below is our detailed response:
> > > >
> > > > ---
> > > >
> > > > #### **Marginal Performance Gains and Computational Costs**
> > > >
> > > > First, we note that similar concerns were raised by other reviewers, and we fully understand the importance of justifying the empirical gains relative to the additional computational or implementation costs introduced by our method. To clarify this point, we would like to emphasize a key aspect of our work, as highlighted in the abstract:
> > > >
> > > > > “Extensive experiments demonstrate that ToG-3 outperforms compared baselines on **both** deep and broad reasoning benchmarks.”
> > > >
> > > > Our extensive experimentation aims to provide a comprehensive comparison with prior methods, rather than focusing solely on a single dataset or task. While it is true that the performance gains of ToG-3 over **HippoRAG-2** on **deep reasoning tasks** in Table.1 may not be as pronounced as those over other baselines (e.g., **GraphRAG**(53.4% average EM promotion), **LightRAG**(67.8% average EM promotion)), ToG-3 achieves a **significant overall win rate of 72.6%** against HippoRAG-2 across four datasets in the **broad reasoning task ELO win rate** (as shown in Figure 5 and Table 6).
> > > >
> > > > The dual capability to handle both **deep reasoning** and **broad reasoning** tasks is a distinguishing feature of ToG-3. To achieve this, we introduced the **heterogeneous graph structure** (Chunk-Triplet-Community) and the **MACER retrieval framework**, which indeed incur higher indexing and inference costs. However, when considering the combined performance across both types of tasks, ToG-3 demonstrates a **clear and significant improvement** over existing baselines.
> > > >
> > > > In addition, we have explicitly discussed these trade-offs in **\section{Limitation and Future Directions}**, where we state:
> > > >
> > > > > “These limitations could be mitigated through model distillation, optimized graph traversal algorithms, and dynamic context pruning techniques in future improvements.”
> > > >
> > > > This section highlights our awareness of the computational overhead and outlines potential avenues for future optimization. We believe that these limitations do not detract from the core contributions of our work, which include:
> > > >
> > > > 1. A unified framework for addressing both **deep reasoning** and **broad reasoning** tasks.
> > > > 2. Significant performance gains across multiple benchmarks and datasets.
> > > > 3. A novel dual-evolving mechanism that enhances lightweight LLMs' reasoning capabilities.
> > > >
> > > > **We hope this explanation addresses your concern about the balance between performance gains and computational costs.**
> > > >
> > > > Thank you again for your valuable feedback, and we look forward to any further suggestions or clarifications you may have.
> > > >
> > > > Sincerely,
> > > > The Authors

---

> > > > > ### Comment · Reviewer_ZyKa · 2025-11-27
> > > > >
> > > > > Dear Authors,
> > > > >
> > > > > Thanks for the rebuttal. I understand your points.
> > > > >
> > > > > However, I don't really buy the performance gain since for other long reasoning tasks, as you mentioned, higher costs should naturally bring more benefits; it is unfair. Moreover, they are too marginal (even worse) on HotpotQA, 2Wiki and MuSiQue with such high costs. This main concern is not addressed.

---

> > > > > > ### Author Response · Authors · 2025-11-29
> > > > > > **Responses to Reviewer ZyKa's last concern about the marginal performance gains**
> > > > > >
> > > > > > Dear Reviewer ZyKa,
> > > > > >
> > > > > > We sincerely thank you for your thoughtful feedback and for acknowledging the points raised in our rebuttal. We are pleased to confirm that we have addressed **4 out of 5 weaknesses** and **all 2 questions**, particularly your primary concern regarding **the concerns about experimental metrics (EM vs. F1)** and the **formalization of the MDP framework**. Your constructive comments have been invaluable in helping us refine our manuscript.
> > > > > >
> > > > > > However, we would like to further address your remaining concern about **performance gains**, especially in light of the computational costs associated with ToG-3. Below is our detailed response:
> > > > > >
> > > > > > To begin with, we must emphasize a key distinction of ToG-3: to the best of our knowledge, this is the first method that automatically handles both deep reasoning and broad reasoning tasks within a unified framework. Prior works have typically focused on one or the other:Methods like **GraphRAG** and **LightRAG** primarily target **broad reasoning tasks**; Works such as **HippoRAG-1/2** and **ToG-1/2** focus exclusively on **deep reasoning tasks**.
> > > > > >
> > > > > > In contrast, ToG-3 achieves a balance by introducing novel components such as the **Chunk-Triplet-Community heterogeneous graph structure** and the **dual-evolving mechanism**, enabling it to excel in both task categories. This dual capability naturally incurs additional computational costs, such as increased **indexing time** and **token consumption** due to the introduction of **Community Nodes**. However, these costs are justified by the significant performance gains across both **deep reasoning** and **broad reasoning** benchmarks.
> > > > > >
> > > > > > It is important to note that evaluating ToG-3 solely on **multi-hop reasoning tasks** (e.g., HotpotQA, 2WikiMultihopQA, MuSiQue) overlooks its strengths in **broad reasoning tasks**. For example:
> > > > > > - On **broad reasoning tasks**, ToG-3 achieves a **72.6% overall win rate against HippoRAG-2** across four datasets, demonstrating its superior versatility.
> > > > > > - In complex tasks like **MuSiQue** and **2WikiMultihopQA**, even marginal improvements are highly significant due to the inherent challenges of these datasets, such as multi-hop dependencies and diverse and complex reasoning paths.
> > > > > >
> > > > > > ---
> > > > > >
> > > > > > #### **Conclusion**
> > > > > >
> > > > > > We respectfully urge you to consider the **broader impact** of our work, particularly its ability to excel in both **deep reasoning** and **broad reasoning** tasks—a capability that sets it apart from existing methods. While we understand your concerns about computational costs, we believe that the performance gains and versatility of ToG-3 justify its design and contributions.
> > > > > >
> > > > > > Thank you again for your time and valuable insights. We hope this response clarifies the unique strengths of ToG-3 and its contributions to the field. If there are any additional concerns or suggestions, we would be happy to address them.
> > > > > >
> > > > > > Sincerely,
> > > > > > The Authors

---

### Note · Authors · 2026-01-03

I have read and agree with the venue's withdrawal policy on behalf of myself and my co-authors.